# Structures of *Helicobacter pylori* C-terminal protease CtpA reveal a new mode of self-contained proteolytic processing

Kailei Sun [1], Lili Yan[1], Chin Yu Mok [1], Pui Kin So[2], Sirius Pui Kam Tse[2], Kwok Fai Lau[3], Daping Wang [4], Huawei Zhang [5,6] ✉ & Shannon Wing Ngor Au [1] ✉

Bacterial carboxyl-terminal processing proteases (CTPs) are essential for protein quality control, signal transduction, and cell adaptation. Typically, activation of CTPs involves substrate binding to the PDZ domain or an adaptor protein. Recent studies of CTPs from various bacterial species have indicated structural diversity in CTP oligomerisation. However, the activation mechanisms and rationale for these different oligomeric forms are not well characterised or understood. Here, we present biophysical analyses of CtpA from *Helicobacter pylori*, which assembles into a trimer-of-dimer (hexameric) configuration. Hydrogen–deuterium exchange mass spectrometry shows that CtpA transitions between resting and active states independently of substrate binding. Cryo-electron microscopy and crystal structural analysis of CtpA further reveal that only one subunit per dimer is active at a time, driven by asymmetric conformational changes. This asymmetric activation supports a cooperative mechanism in which hexameric subunits synchronise to regulate proteolytic activity. Coordinated inter-subunit interactions and concerted movements of the PDZ domain and motile loop generate three self-compartmentalised catalytic units that enable processive substrate degradation. We also identify intra- and intermolecular interactions that stabilise functional states, allowing adaptor-independent activation. These findings open a new avenue towards understanding the key elements of oligomeric assembly in protease activation.

Carboxyl (C)-terminal processing proteases (CTPs) are ATP-independent serine proteases that mediate proteolytic cleavage at the C-termini of substrate proteins. These proteases occur in all domains of life and are crucial for protein quality control and signal transduction[1–3]. In bacteria, CTPs regulate key physiological processes such as cell wall remodelling, stress responses and cell cycle progression[4–9]. Although the protein sequence of a CTP can vary from approximately 400–700 amino acid residues in length, sequence homology is used to classify bacterial CTPs into three subfamilies: CTP-1, CTP-B and CTP-3[10]. Studies of representative structures in each

subfamily, such as Prc in *Escherichia coli*, CtpB in *Bacillus subtilis* (*Bs*CtpB) and CtpA in *Pseudomonas aeruginosa* (*Pa*CtpA)[9,11,12], have revealed diverse activation mechanisms.

One major function of bacterial CTPs is the modulation of peptidoglycan (PG) metabolism. In *E. coli*, Prc works with the adaptor NlpI to degrade hydrolases, such as MepS, thus ensuring proper remodelling of the PG layer during growth and division[4,11]. In *P. aeruginosa*, a parallel system involving *Pa*CtpA and lipoprotein LbcA targets hydrolases such as MepM and PA1199, underscoring a conserved strategy in bacterial envelope

¹Center for Protein Science and Crystallography, School of Life Sciences, The Chinese University of Hong Kong, Shatin, Hong Kong, China. ²State Key Laboratory of Chemical Biology and Drug Discovery, Department of Applied Biology and Chemical Technology, The Hong Kong Polytechnic University, Kowloon, Hong Kong, China. ³School of Life Sciences, The Chinese University of Hong Kong, Shatin, Hong Kong, China. ⁴Department of Orthopedics, Shenzhen Intelligent Orthopaedics and Biomedical Innovation Platform, Guangdong Artificial Intelligence Biomedical Innovation Platform, Shenzhen Second People's Hospital, the First Affiliated Hospital of Shenzhen University Health Science Center, Shenzhen, China. ⁵Shenzhen Institute of Advanced Technology, Chinese Academy of Sciences, Shenzhen, China. ⁶School of Life Sciences, Southern University of Science and Technology, Shenzhen, China. ✉e-mail: hw.zhang@siat.ac.cn; shannon-au@cuhk.edu.hk

regulation[5]. *Bs*CtpA additionally contributes to DNA damage recovery in *B. subtilis* by degrading the checkpoint inhibitor YneA, thus facilitating cell division post-repair[8]. CTP-null strains in *Brucella suis*, *Burkholderia mallei* and *Staphylococcus aureus* all exhibit reduced virulence[13–15], highlighting the importance of these proteases in bacterial pathogenesis and their potential as therapeutic targets. Despite these insights, our understanding of CTP function is incomplete due to the limited number of substrates identified.

A typical bacterial CTP consists of an N-terminal domain, a PDZ domain, a protease domain (further subdivided into cap and core sub-domains) where the catalytic triad resides, and a C-terminal domain (Fig. 1a)[9,11,12]. Despite the highly flexible orientation of the substrate-binding PDZ domain, the PDZ-protease unit is the most structurally conserved, whereas the N- and C-terminal domains participating in intermolecular interactions that generate different CTP oligomeric forms are variable[9,16,17]. For example, *Bs*CtpB, *Pa*CtpA, and *Acinetobacter baumannii* S41 peptidase dimerise by packing their N-terminal helices into a four-helix bundle[9,12,16]. The *Pa*CtpA dimer further trimerises via the trans-interacting loop and long helix (C-helix) located at the C-terminal of each subunit to form a hexamer[12,16]. Prc remains monomeric, and its N- and C-terminal β-strands associate to create a bowl-shaped structure[17], with the PDZ domain acting as a regulatory switch[9,17,18]. In the resting state, this domain docks over the protease tunnel, thus misaligning the catalytic residues, while in the active state, it moves outward and realigns the active site for cleavage (Fig. 1b). Prc requires its PDZ domain for activation[11,17], whereas the PDZ domain of *Bs*CtpB is autoinhibitory, such that its deletion yields a protease with constitutive activity[9].

Recent studies have observed that membrane adaptor proteins may also regulate CTP activity. The crystal structure of Prc in complex with the adaptor NlpI and substrate MepS shows that NlpI facilitates Prc activity by bringing MepS and Prc together[11,19]. By contrast, the adaptor LbcA not only acts as a medium between *Pa*CtpA and its substrates but also regulates *Pa*CtpA activity by stabilising its hexamer[12,16]. Taken together, these observations suggest that bacterial CTPs have evolved to use different molecular designs for enzyme activation and regulation. Understanding how these structural differences influence CTP behaviour is essential for gaining deeper insights into their functional roles.

In this work, we combine protein crystallography, cryo-electron microscopy (cryo-EM) and hydrogen–deuterium exchange mass spectrometry (HDX-MS) to study the structural basis of *Helicobacter pylori* CtpA (*Hp*CtpA), a member of the CTP-3 subfamily. *H. pylori* is the primary pathogen colonising human stomachs and its presence strongly correlates with the incidence of gastric cancer[20]. CtpA is one of the least well-characterised of the 20 proteases encoded in the *H. pylori* genome[21]. Its importance and pathological relevance are indicated by its high sequence conservation and detection of the *ctpA* gene in a majority of clinical isolates[22]. Here, we present the proteolytic mechanism of *Hp*CtpA with respect to the putative substrate HP1076, which acts as a co-chaperone of the flagellin export chaperone FliS[23] and is involved in bacterial motility[24]. Our structural and dynamic data demonstrate that *Hp*CtpA is a conformationally dynamic protease that undergoes structural transitions for intrinsic auto-activation and exhibits a processive nature in trimer-of-dimer (i.e., hexameric) assembly. These findings exemplify a new mechanism of CTP self-assembly for effective proteolysis.

## Results and discussion
### Structures of *H. pylori* CtpA

*H. pylori* CtpA was first identified in a pull-down assay conducted to find interacting partners of HP1076 (Supplementary Fig. 1a). Compared with the control experiments using GST-FliS and GST as bait (lanes 4 and 5), two additional bands were observed in the GST-FliS-HP1076 pull-down product (lane 6). These two bands were subjected to mass spectrometry analysis, which identified them as CtpA and cleaved HP1076, with the C-terminal 21 amino acid residues undetected. Given that CtpA functions as a C-terminal processing protease, we next studied the proteolytic activity of CtpA towards HP1076. To ensure the intracellular expression of recombinant CtpA using an *E. coli* expression system, we deleted a 20-amino-acid sequence from the N-terminus, which was predicted to be a signal peptide, from the construct. However, the resulting protein product was unstable (Supplementary Fig. 1b). Thus, multiple N-terminal truncated CtpA mutants were screened. Among these, as the most stable construct retaining the longest N-terminal end, CtpAΔN46 (residues 47-454) was selected (hereafter referred to as CtpA or *Hp*CtpA) and used for subsequent investigations in this study. From the results of the CtpA activity assays, a cleaved product of HP1076 was detected over time only in the presence of wild-type CtpA (Supplementary Fig. 1c). The inactive mutant CtpA$_{S300A}$ (substitution of serine at residue 300 with alanine)[9,11,12], which was set up in parallel produced no cleaved band, indicating that the serine at residue 300 is critical for CtpA's cleavage activity. The direct interaction of CtpA with HP1076 was further confirmed using MST, which showed a Kd of 15.4 μM (Supplementary Fig. 1d). Moreover, both CtpA and HP1076 were detected in the membrane fraction; while CtpA is membrane-bound, HP1076 appears to be predominantly membrane-associated, although it is also

**Fig. 1 | Schematic diagram of a typical CTP. a** The domain organisation of CtpA in *H. pylori*. The residue range of each domain and the positions of the catalytic triad are marked. **b** The resting CTP (left) and active CTP (right) are depicted with the different domains marked.

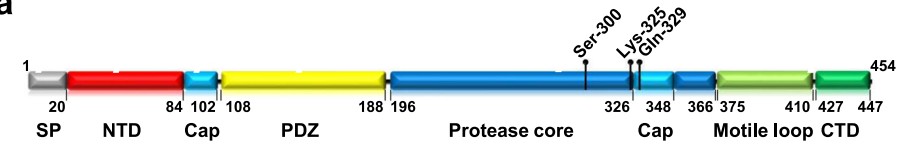

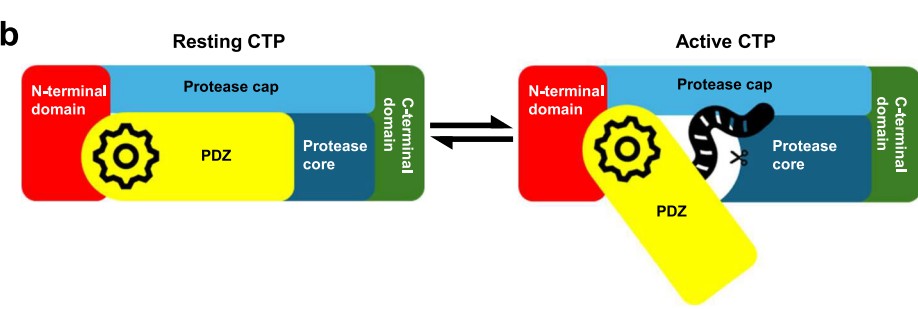

## Table 1 | Crystallography data collection and refinement statistics

| | CtpA PDB 9JR1 | CtpA$_{F105R}$ PDB 9KU0 |
|---|---|---|
| **Data collection** | | |
| Space group | P1211 | H3 |
| Cell dimensions | | |
| $a, b, c$ (Å) | 61.7, 229.1, 116.6 | 139.3, 139.3, 252.1 |
| $\alpha, \beta, \gamma$ (°) | 90.0, 101.3, 90.0 | 90.0, 90.0, 120.0 |
| Resolution (Å)[a] | 27.1–3.4 (3.5–3.4) | 28.0–3.7 (4.1–3.7) |
| No. of unique reflections[a] | 41,187 (2714) | 19,454 (4677) |
| $R_{merge}$ (%)[a] | 8.4 (49.2) | 12.6 (52.1) |
| $I / \sigma I$[a] | 14.3 (2.0) | 7.9 (3.1) |
| CC1/2 (%)[a] | 99.0 (70.8) | 99.4 (84.7) |
| Completeness (%)[a] | 94.6 (63.1) | 99.8 (100.0) |
| Redundancy[a] | 3.4 (2.4) | 5.2 (5.2) |
| **Refinement** | | |
| $R_{work} / R_{free}$ (%) | 19.3/25.0 | 24.3/31.1 |
| No. atoms | | |
| Protein | 17,123 | 11,012 |
| Ligand/ion | 0 | 0 |
| Water | 0 | 0 |
| Average B-factor (Å²) | 60.4 | 142.9 |
| R.m.s. deviations | | |
| Bond lengths (Å) | 0.003 | 0.005 |
| Bond angles (°) | 0.683 | 0.880 |
| Ramachandran plot (%)[b] | 93.9/5.1/1.0 | 94.1/5.8/0.1 |
| Clash score | 4.1 | 7.7 |

[a] Values in parentheses are for highest resolution shell.
[b] Number of residues in favoured region/allowed region/outlier region.

localised in the cytosol (Supplementary Fig. 1e). These results showed that HP1076 is a putative substrate of CtpA.

Analytic ultracentrifugation (AUC) revealed that CtpA is a hexamer in solution (Supplementary Fig. 1f) and presents as a triangular ring structure as revealed by transmission electron microscopy (TEM) (Supplementary Fig. 1g). AUC also identified a minor species with a molecular weight of 474 kDa, which may represent residual protein impurities or aggregates of CtpA formed during ultracentrifugation. However, TEM did not reveal any higher-order oligomeric assemblies. We further obtained the crystal structure of CtpA at a resolution of 3.4 Å (Table 1). CtpA has a hexameric ring or trimer-of-dimer structure (Supplementary Fig. 2a), resembling that of *Pa*CtpA. An N-terminal dimerisation domain (NTD) and C-terminal dimerisation domain (CTD) flanking the protease core are responsible for hexamerization. Regarding the PDZ domains, three with clear electron densities are compact and located at one face of the hexamer. The remaining three PDZ domains, with weak densities, are located on the opposite face and shifted away from the main body. Of these, two were bound to substrate peptides presumably co-purified from *E. coli*. Based on the orientations of the PDZ domains and the presence of substrate peptides, the CtpA crystal structure was determined to contain three subunits in the resting state, two subunits in the active state and one subunit in an intermediate state (Supplementary Fig. 2b–e). The α3 belonging to the protease cap subdomain is flexible in the intermediate subunit structure. In each subunit, about 30–40 residues upstream of the C-helix remain unassigned.

To reduce the structural flexibility induced by substrate proteolysis, a triple mutant CtpA$_{S300A/K325A/Q329A}$ (CtpA$_{TM}$) with a disabled catalytic triad[9,11,12] was generated. A structural model of CtpA$_{TM}$ was built using a cryo-EM map with a resolution of 3.13 Å (Supplementary Fig. 3 and 4, Table 2). Each face of the CtpA$_{TM}$ hexamer comprises three subunits in the

same state, either active or resting (Fig. 2a). For the active face side of the hexamer, a substrate peptide modelled with poly-alanine was assigned next to the shifted PDZ domain (Fig. 2b, c). A structural comparison of the resting and active subunits revealed that, along with the shift in the PDZ domain, the hinge region (residues 104–107 and 189–192) connecting the PDZ domain to the cap and core subdomains is transformed from loops to a β-sheet (Fig. 2c). Furthermore, the active subunits were found to exhibit other conformational changes, including movement of the NTD and cap subdomain as described in *Bs*CtpB and *Pa*CtpA[9,12,16]. Residues 376–392 are modelled between the core subdomain and the CTD (Fig. 2c). This segment and residues 393–422, which are predicted to be disordered, constitute the motile loop (ML), so named for its flexible position within the active and resting subunits. Specifically, ML either contacts the PDZ domain or inserts into the cleft between an active and a resting subunit. As described in later sections, this ML region helps to regulate CtpA activity.

### Structural dynamics of CtpA

The co-existence of resting and non-resting states revealed in the CtpA crystal structure and CtpA$_{TM}$ cryo-EM structure suggests that the protease switches between these two states by swinging the PDZ domain inwards and outwards. We used HDX-MS to investigate the intrinsic dynamics of CtpA. Similar studies of CTPs have not been documented previously. The CtpA$_{TM}$ construct was denatured with urea, refolded and purified to remove the endogenously bound substrate from *E. coli*, yielding apo CtpA$_{TM}$. In total, 91.1% of the CtpA sequence was covered by the MS (Supplementary Fig. 5a and Supplementary Table 1). For each subunit within the CtpA hexamer, limited deuterium uptake was observed near the active site (residues 304–320) and the C-terminal dimerisation interface (residues 430–438) even after 2 h of labelling, indicating the rigidity of these structures

**Table 2 | Cryo-EM data collection, refinement, and validation statistics**

| | CtpA$_{TM}$<br>EMD-62573<br>PDB 9KU3 | CtpA conformation I<br>EMD-62575<br>PDB 9KUB | CtpA conformation II<br>EMD-62576<br>PDB 9KUC | CtpA conformation III<br>EMD-62553<br>PDB 9KSP |
|---|---|---|---|---|
| **Data collection and processing** | | | | |
| Magnification | 130k | 105 K | 105 K | 105 K |
| Voltage (kV) | 300 | 300 | 300 | 300 |
| Electron exposure (e–/Å2) | 50 | 50 | 50 | 50 |
| Defocus range (μm) | -1.5 to -2.5 | -1.5 to -2.5 | -1.5 to -2.5 | -1.5 to -2.5 |
| Pixel size (Å) | 0.92 | 0.83 | 0.83 | 0.83 |
| Symmetry imposed | C3 | C3 | C1 | C1 |
| Initial particle images (no.) | 1,449,887 | 561,739 | | |
| Final particle images (no.) | 151,917 | 155,703 | 133,262 | 144,375 |
| Map resolution (Å) | 3.13 | 3.43 | 3.59 | 3.49 |
| FSC threshold | 0.143 | 0.143 | 0.143 | 0.143 |
| Map resolution range (Å) | 2.84–11.06 | 2.81–8.07 | 2.93–10.15 | 2.89–12.33 |
| **Refinement** | | | | |
| Model resolution (Å) | 3.39 | 3.52 | 3.79 | 8.15 |
| FSC threshold | 0.5 | 0.5 | 0.5 | 0.5 |
| Map sharpening $B$ factor (Å$^2$) | -155.2 | -130.8 | -121.7 | -128.2 |
| Model composition | | | | |
| Non-hydrogen atoms | 17,769 | 16,116 | 17,214 | 15,319 |
| Protein residues | 2317 | 2098 | 2233 | 1993 |
| Ligands | 0 | 0 | 0 | 0 |
| R.m.s. deviations | | | | |
| Bond lengths (Å) | 0.012 | 0.003 | 0.002 | 0.003 |
| Bond angles (°) | 1.826 | 0.560 | 0.529 | 0.610 |
| Validation | | | | |
| MolProbity score | 0.84 | 1.88 | 1.72 | 2.22 |
| Clashscore | 0.11 | 6.28 | 5.06 | 9.42 |
| Poor rotamers (%) | 0.98 | 1.95 | 0.68 | 3.64 |
| Ramachandran plot | | | | |
| Favoured (%) | 95.72 | 95.47 | 92.98 | 95.69 |
| Allowed (%) | 4.28 | 4.53 | 7.02 | 4.31 |
| Disallowed (%) | 0.00 | 0.00 | 0.00 | 0.00 |

(Supplementary Figs. 5b and 6). Conversely, the surface solvent-exposed area showed significant deuterium uptake within a short period, in line with the structural model. Bimodal distribution patterns were observed, especially for the peptides covering α3 in the protease cap subdomain and the hinge connecting the PDZ domain (residues 90–113) and part of the ML (residues 393–422) (Fig. 3a, b). This bimodal distribution pattern may indicate the presence of multiple conformations[25]. It is plausible that CtpA subunits exist in equilibrium between the active and resting states in solution, independently of substrate binding. We speculate that the resting form is dominant because it has a larger interface area between the PDZ domain and the main body of CtpA than that of the active form (777.4 vs 363.5 Å$^2$). Next, to monitor the molecular dynamics of CtpA during substrate binding, the apo CtpA$_{TM}$ sample was incubated with an excess of HP1076 to shift the equilibrium of the hexamer subunits to the active conformation to the greatest extent possible. Complexing CtpA with HP1076 showed a protective effect of deuterium uptake at multiple sites, including α2 of the NTD (residues 73–82), part of the PDZ domain (residues 105–142), part of the protease core subdomain (residues 238–256) and the ML contacting region containing residues 345–364 (Fig. 3c and Supplementary Fig. 5c, d). The regions exhibiting decreases in deuterium uptake are likely to be related to the interfaces that come in contact with the substrate HP1076 and/or the

structural changes induced upon HP1076 binding. These include the tunnel surrounded by the protease cap and core subdomains, the PDZ domain, the hinge and the region in the activated protease core subdomain near the ML. The decreases in deuterium uptake also suggest that, apart from the PDZ domain, which reaches outwards, the remainder of active CtpA has a more compact and rigid structure that provides additional protection from deuteriation of the NTD and cap subdomains. The bimodal spectral distribution of α3 and the ML region was also found in the CtpA-HP1076 sample (Supplementary Fig. 7), indicating that when CtpA was saturated by the substrate, the structure still contained a combination of subunits with different conformations. This finding is consistent with the crystal and cryo-EM structures, in which half of the subunits in one hexamer were found to exist in the resting state. These results suggest a potential structural constraint that prevents all subunits in the hexamer from shifting to the active state simultaneously.

**Asymmetric conformational changes and assembly of the self-compartmentalisation unit in CtpA**

The crystal structure of CtpA and the cryo-EM structure of CtpA$_{TM}$ both reveal three resting subunits and three non-resting subunits located at opposite faces of the hexamer, and this subunit arrangement led us to

**Fig. 2 | Cryo-EM structure of the CtpA$_{TM}$ mutant.**
**a** Cryo-EM density map of the CtpA$_{TM}$ hexamer.
The resting and active subunits are indicated by cold
and warm colours, respectively. The substrates are
coloured in yellow. **b** Structure of CtpA$_{TM}$ in a
cartoon presentation. The subunits are coloured as
in (**a**). The substrates are depicted in molecular
surface mode. **c** Enlarged view showing the differ-
ential conformations relative to the protease core
domain in the resting state (left) and active state
(right). The domains are coloured as depicted in
Fig. 1a, and the substrate is coloured in brown with a
transparent molecular surface. The motile loops are
displayed as thick ribbons.

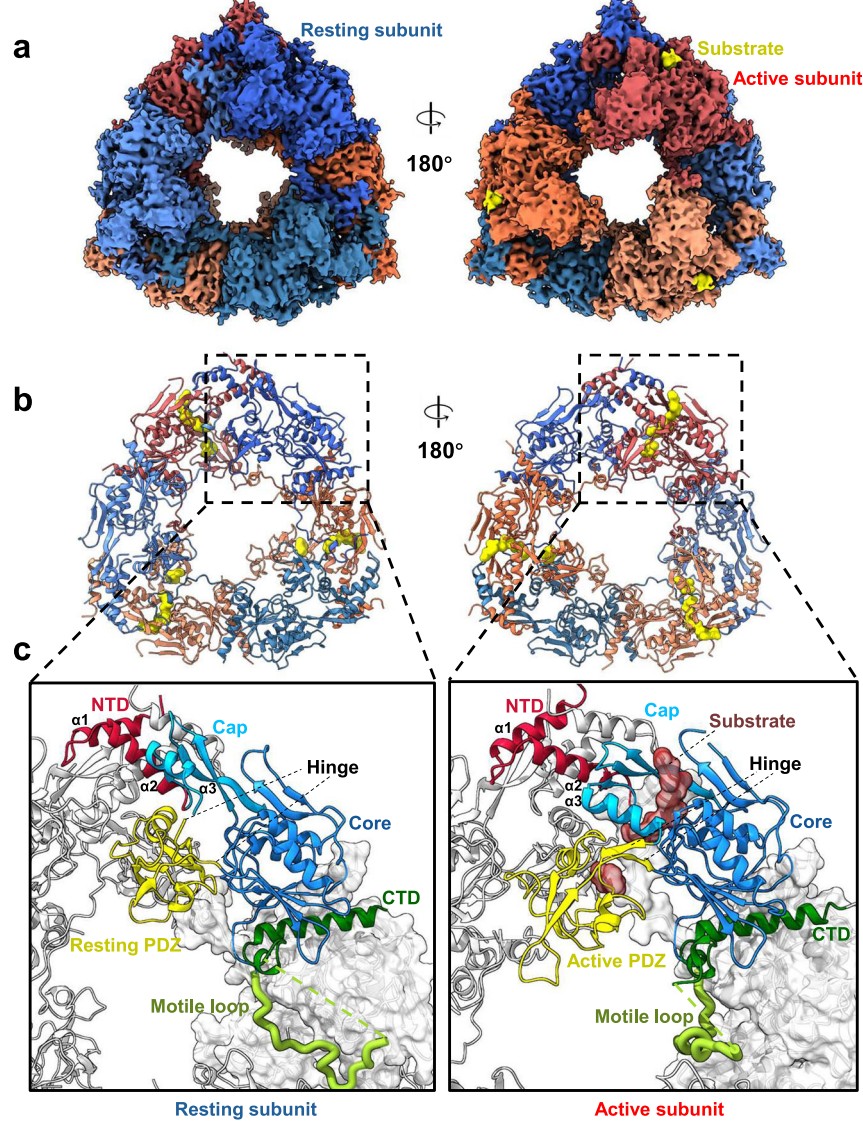

explore the regulatory mechanism of CtpA activation. We attempted to
visualise the dynamic conformational changes in wild-type CtpA using
cryo-EM. Heterogeneous refinement resulted in three maps corresponding
to CtpA in different conformations: (I) three resting subunits at one face and
three non-resting subunits at the other face (map resolution: 3.43 Å); (II)
two resting subunits and one non-resting subunit at one face, and one
resting subunit and two non-resting subunits at the other face (map reso-
lution: 3.59 Å); and (III) six resting subunits (map resolution: 3.49 Å)
(Fig. 4a). The differential arrangement of the resting and non-resting sub-
units within a CtpA hexamer implies that at least one subunit within a CtpA
dimer mediated by the NTD (N-dimer) is in the resting state. On this basis, a
maximum of three subunits within a CtpA hexamer would be active. This
conclusion is consistent with the *Hp*CtpA structures revealed in this study
and the currently available *Pa*CtpA structures[12,16].

This phenomenon of at least one resting state per N-dimer may be
related to the mechanism of CtpA subunit activation. A detailed structural
analysis revealed that extensive hydrogen bonds and salt bridges are formed
between the two NTDs and protease cap subdomains (excluding α3) of each
subunit in a single N-dimer. This group of interactions form a rigid and
isolated module (referred to as the dynamic unit) (Fig. 4b, Supplementary
Fig. 8 and Supplementary Table 2). At the two distal ends of the N-dimer, the
protease core subdomain and the CTD constitute another rigid module

referred to as static units 1 and 2 (Fig. 4b). Within a single N-dimer, con-
straints from the C-terminal dimerisation interfaces maintain the ring
structure. Therefore, the positions of static units 1 and 2 are largely fixed,
leaving the dynamic unit to shift towards either static unit 1 or 2. This shift is
coupled with the outward movement of the corresponding PDZ domain in
the active subunit. Consequently, an asymmetric conformational change
happens within one N-dimer in which only one subunit within the N-dimer
can be activated, while the other subunit necessarily remains at rest (Fig. 4b).
Following this transformation, the protease tunnel is exposed in the acti-
vated subunit for substrate binding. Although *Bs*CtpB shares an N-terminal
dimerisation pattern with *Hp*CtpA and *Pa*CtpA, its C-terminal dimerisa-
tion is facilitated by a four-helix domain positioned between the protease
domain and the C-helix, rather than the long C-helix used in *Hp*CtpA and
*Pa*CtpA (Supplementary Fig. 9). This difference in dimerisation interface
architecture not only results in a transition of the oligomeric state from
hexameric to dimeric but also enables both subunits within *Bs*CtpB to be
independently active. Therefore, we speculate that the C-dimerisation
interface of *Bs*CtpB does not generate structural constraints. These findings
imply that switching of the PDZ domain can be regulated by other structural
modules.

The asymmetric conformational change observed in each N-dimer
would reduce the catalytic power of CtpA by half. We recognised this

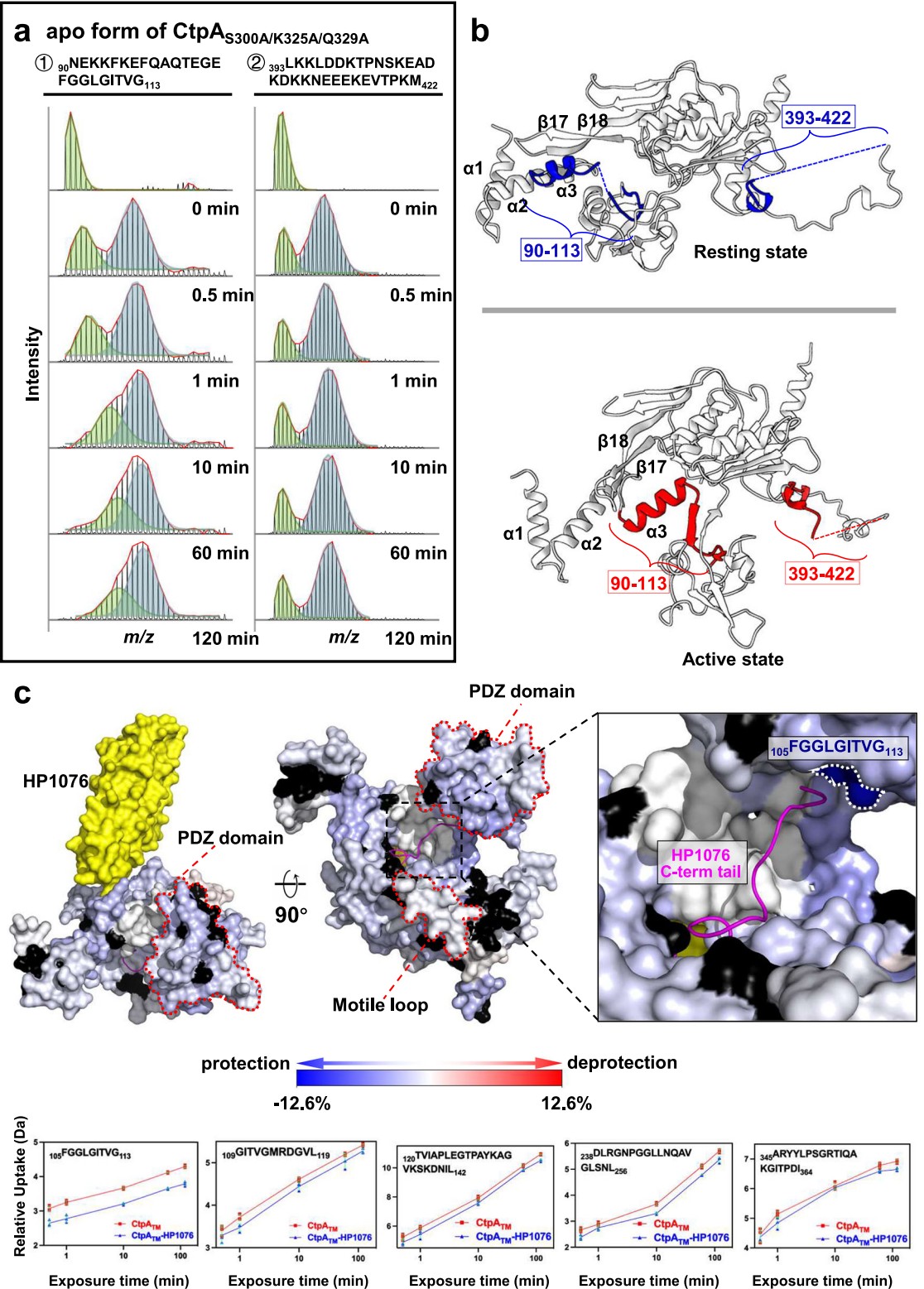

**Fig. 3 | Structural dynamics of CtpA. a** HDX-MS stacked spectral plots of peptides covering residues 90–113 and 393–422 from apo CtpA_{TM}. The HDX intervals were 0, 0.5, 1, 10, 60, and 120 min. The bimodal distribution (green and blue peaks) indicates the simultaneous presence of multiple states. **b** The peptides in (**a**) are mapped in the resting state (blue; upper panel) and active state (red; lower panel) of the CtpA structure. **c** Docking of HP1076 to the CtpA subunit bound to the substrate peptide. The summed HDX difference of 10- and 120 min labelling is mapped on the CtpA structure. Unidentified peptides in HDX-MS are coloured in black. Enlarged view shows the contact interface between the C-terminal tail of HP1076 and CtpA. The sequence of the peptide with the largest HDX difference is labelled. Kinetic plots of peptides with a significant ΔHDX are shown at the bottom, with error bars representing the standard deviation ($n = 3$ independent experiments).

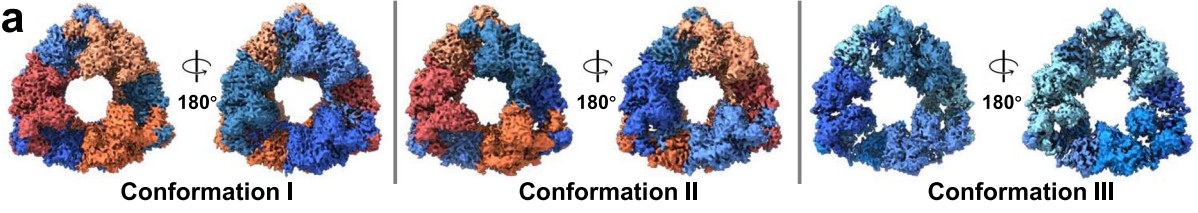

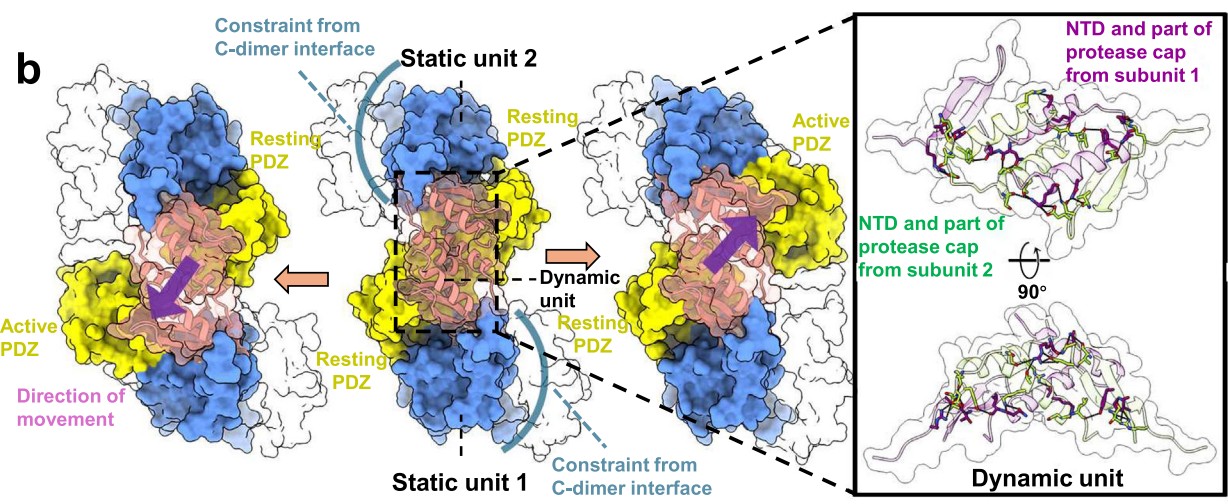

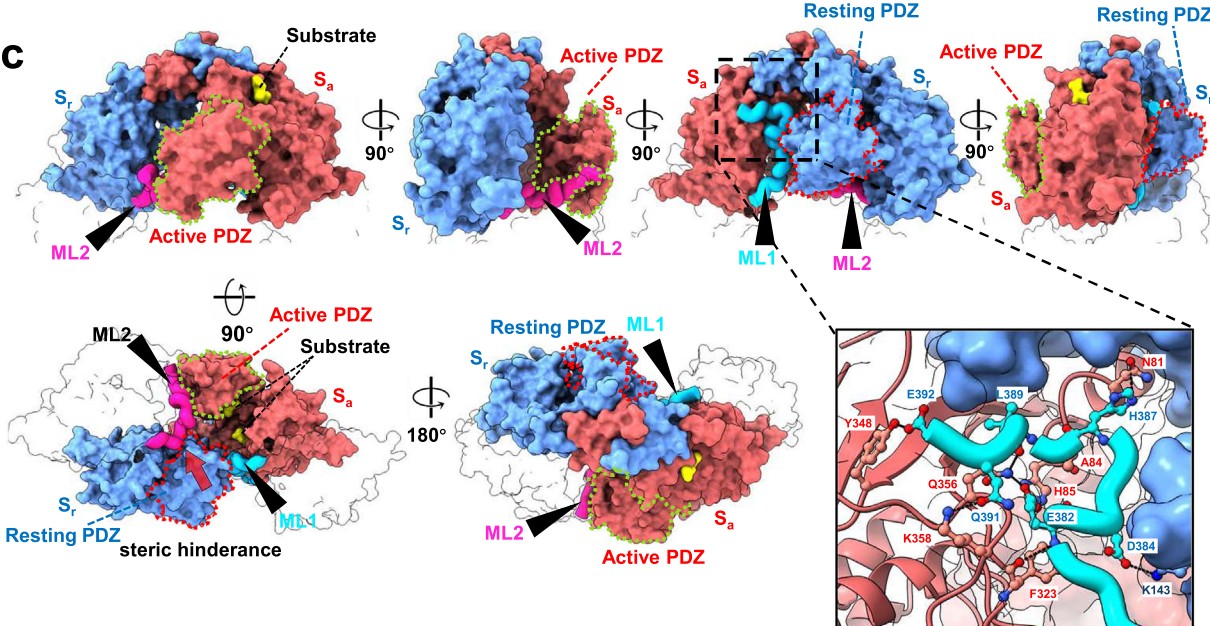

**Fig. 4 | Self-compartmentalisation of CtpA. a** Cryo-EM density maps of three particle populations of wild-type CtpA identified via heterogeneous refinement. Resting and non-resting subunits are labelled with cold and warm colours, respectively. **b** Surface view of an N-terminal dimer showing the asymmetric conformation. The dynamic unit is depicted transparently with secondary structures. Static units 1 and 2 are coloured in blue, and the PDZ domains are coloured in yellow. Each static unit moves to either side as indicated by the purple arrows. The enlarged view shows hydrogen bonds within the dynamic unit. The two involved subunits are shown as semi-transparent ribbon models, coloured green and purple, respectively.

**c** A self-compartmentalising working unit presented in surface view. The active ($S_a$) and resting ($S_r$) subunits are coloured in red and blue, respectively. The PDZ domains are indicated by dotted lines. The motile loops provided by the C-dimer partners are depicted as ribbons and indicated as arrowheads. These two C-dimer partners are presented transparently. The substrate is coloured in yellow. The enlarged window illustrates the ML1 binding interface. Residues involved in hydrogen bond and salt bridge formation are marked. Steric hinderance from the resting PDZ domain that repels ML2 is marked by a red arrow.

**Fig. 5 | Stabilisation of the functional and oligomeric states of CtpA. a** The substrate binding site in an active subunit. The enlarged view shows how the stabilisation of the active state is mediated by the salt bridge between Glu-96 and Arg-162 and the cation–π interaction between Phe-105 and Arg-162 (dotted box). Processive proteolysis of the C-terminus of the substrate is driven by the protonated Arg-162. The cryo-EM density near the C-terminus of the substrate peptide is shown. **b** The crystal structure of the CtpA$_{F105R}$ mutant, displayed in surface view. The six resting subunits are labelled in cold colours, and one of the subunits is transparent. **c** The crossing angles of the two α-helices in the C-terminal dimerisation interface of CtpA from *P. aeruginosa* (left) and *H. pylori* (right).

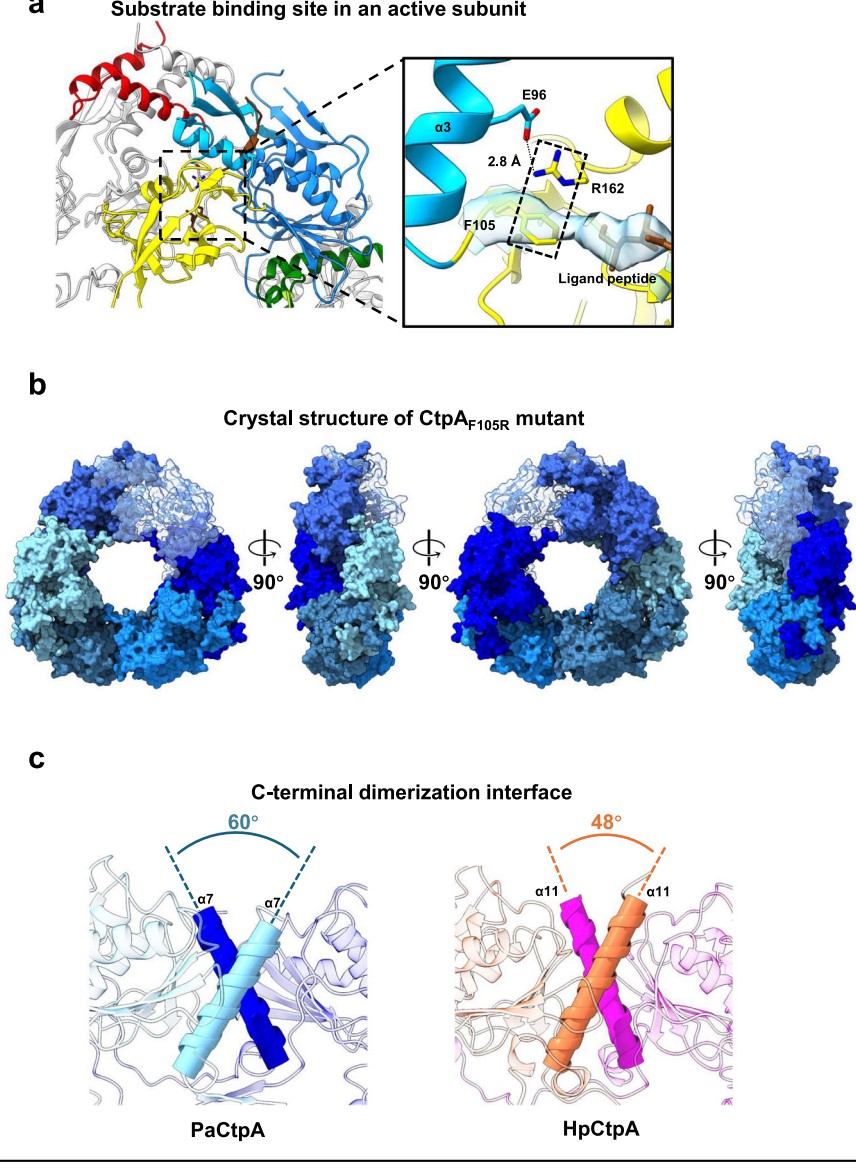

necessary change when we examined the structural details of the N-dimers and their respective C-dimers (i.e., the CtpA dimer mediated by the CTD) in both CtpA$_{TM}$ and CtpA in conformations I and II. We found that each N-dimer and the two MLs (ML1 and ML2), each from a C-dimer partner, assemble to form a self-compartmentalising protease unit (Fig. 4c). Regarding the active subunit (S$_a$), its narrow protease tunnel could be accessed via either face of the hexamer. However, substrate entrance at one face is blocked by the PDZ domain of the resting subunit (S$_r$) and ML1 from the C-dimer partner of S$_a$. Specifically, ML1 fills the gap between the protease core subdomain of S$_a$ and the PDZ domain of S$_r$. The occupancy of ML1 in this region is achieved by a shift of the PDZ domain in S$_a$, which leads to exposure of the NTD and protease core subdomain. This exposure enables extensive hydrogen bonding and salt bridge formation between ML1 and the surrounding region (Fig. 4c, Supplementary Fig. 8 and Supplementary Table 3). In contrast, the PDZ domain of S$_r$ occupies the interface on the NTD and the protease core subdomain of S$_r$, thus repelling ML2 from the C-dimer partner of S$_r$. ML2 moves with the shifted PDZ domain of S$_a$ and partially seals the bottom of the self-contained unit (Fig. 4c). Therefore, the concerted swapping of MLs with subunit activation acts as a gatekeeper to ensure one-way substrate entry in S$_a$. While the specific interaction between ML2 and the PDZ domains of S$_a$ cannot be defined due to the low local resolution, Glu-382 in ML1 is linked to His-85 of

the protease core in S$_a$ via hydrogen bonds (Fig. 4c). CtpA activity was found to be largely reduced in the H85A and E382A mutants and totally abolished in the E382R mutant (Supplementary Fig. 10). In *Pa*CtpA, the equivalent residue His-84 functions as the acid residue of the catalytic triad[16]. Our results further demonstrate the importance of the MLs for full proteolytic function. Both His-85 and Glu-382 are highly conserved among CtpAs but not CtpBs; these residues are likely to have evolved in a pairwise manner for activity regulation (Supplementary Fig. 8).

## ATP-independent processive proteolysis of *Hp*CtpA is mediated by Arg-162

A self-compartmentalising protease can undergo processive proteolysis, during which the substrate is continuously processed, without releasing the intermediate, until digestion is completed[26,27]. The processivity of CtpA towards HP1076 was assayed using reversed-phase high-performance liquid chromatography (HPLC) (Supplementary Fig. 11). The HPLC profiles of the product peaks varied for the non-processive protease trypsin. However, proteolysis by CtpA yielded a decrease in the substrate peak and an increase in the product peaks over time. These results indicate that CtpA undergoes processive digestion. In previous research on the ATP-independent processive protease MtaLonC, an electrostatic ratchet mechanism was proposed in which a protonated Glu attracts the nascent

C-terminus of the substrate, which is translocated along the proteolytic groove in each cleavage cycle[28]. In CtpA, the basic residue Arg-162 at the ligand binding site in the PDZ domain can be readily protonated under the experimental condition of pH 7.5. Therefore, we speculate that Arg-162 acts to attract substrates (Supplementary Fig. 12). The resting PDZ domain and ML1 shield the substrate to prevent its escape in the opposite direction towards the activated PDZ domain and facilitate translocation of the nascent intermediate in the self-compartmentalised unit. Consistent with these observations, HDX-MS revealed that the deuterium exchange of the ML was protected after being bound with HP1076 (Supplementary Fig. 5c).

Self-compartmentalisation and its associated processivity comprise a strategy utilised by proteases to control proteolysis, thus preventing proteins from unintentional degradation. The target substrate must be recognised, unfolded and translocated through a narrow pore to reach the catalytic sites[29]. This process is often ATP-driven, as exemplified by ClpX in ClpXP protease and Lon protease[28]. In H. pylori, four ATP-dependent self-compartmentalising proteases are present in the genome: HslVU, ClpXP, Lon and FtsH[30,31]. These proteases share similar features, namely self-compartmentalisation via protomer oligomerisation into barrel-shaped structures[31]. Here, trimerisation of the dimer in CtpA generates three self-compartmentalisation units, each with an asymmetric conformation. Substrate translocation is presumed to be driven by the protonated Arg-162. These properties allow CtpA to efficiently control its activity in an ATP-free periplasmic environment, at a cost of half of the CtpA molecules.

### Substrate recognition and conformation maintenance of CtpA

The conserved residue Arg-168 in BsCtpB is crucial for substrate sensing and active conformation maintenance[9]. The equivalent residue Arg-162 in CtpA is salt-bridged with Glu-96 in α3 of the protease cap subdomain, where it stabilises the PDZ domain in the $S_a$ subunits (Fig. 5a). Mutants R162A and R162E in CtpA exhibited a loss of enzyme activity, although the effect of R162E was greater (Supplementary Fig. 10). The correct orientation of Arg-162 is likely to be related to the cation–π interaction with a highly conserved Phe-105 in the hinge of the PDZ domain. An electron density connecting the C-terminus of the substrate and the side chain of Phe-105 is observed in CtpA$_{TM}$, highlighting the involvement of Phe-105 in substrate binding (Fig. 5a). In line with these observations, the F105R mutant exhibited drastically impaired activity, whereas the F105Y mutant retained proteolytic activity (Supplementary Fig. 10). The 3.7 Å crystal structure of the F105R mutant revealed a hexamer with all protomers in the resting state, further suggesting that Phe-105 is essential for maintaining the active conformation (Fig. 5b and Table 1). Additionally, HDX-MS analysis of the CtpA$_{TM/F105R}$ mutant revealed monomodal spectral patterns for peptides spanning residues 90–113 and 394–422 (Supplementary Figs. 13 and 14). Comparison with HDX-MS data from the CtpA$_{TM}$ mutant further confirmed that the F105R mutation impaired the conformational transition from the resting to the active state.

Although HpCtpA and PaCtpA are structurally very similar, the former is autoactivated in vitro. In PaCtpA, LbcA must bind to the NTD to activate the protease and enable trimer-of-dimer formation through the CTD[12,16]. Structural comparisons of the two CtpA proteases revealed various insights, leading us to propose that the activation mechanism is species-specific. Although the CTD domains are structurally conserved, the crossing angle of helices α7 in the interface of the PaCtpA apo form is about 12° greater than that in the interface of HpCtpA (Fig. 5c). This results in a smaller interface area (319 Å$^2$) in the PaCtpA C-dimer than in HpCtpA (436 Å$^2$). The interaction between LbcA and the NTD domain allows a particular PDZ domain to shift in the N-dimers and the ML to move inward to promote C-dimer formation. All cap subdomains in the crystal structure of the PaCtpA hexamer are in the resting state[12], further suggesting that PaCtpA does not intrinsically possess an active-resting state equilibrium as seen in HpCtpA. In contrast, the binding interface at the CTD in HpCtpA is sufficient to maintain the hexameric form even if all subunits are in the resting state.

In conclusion, the structural and biophysical characterisations of the self-compartmentalised HpCtpA indicate its molecular dynamics, where only one subunit can be active per N-dimer. The concerted movement of the PDZ domain and the MLs from the C-dimer partners during the conversion between resting and active states only allows the unfolded C-terminus of a substrate to enter the active subunit and be processed in a processive manner. This work also explains the role of conserved Arg-162 and Phe-105 (Supplementary Fig. 8) in interacting with the C-terminus of a substrate in the activation process and opens a new perspective to understand the structural basis of C-terminal proteases. It remains unclear why HpCtpA acts so differently from PaCtpA, and thus, additional biological functional studies are needed.

## Methods

### H. pylori growth conditions

H. pylori G27 strain (kindly provided by Professor Karen Ottemann, Department of Microbiology and Environmental Toxicology, University of California) was cultured at 37 °C on Columbia blood agar with 5% defibrinated horse blood under microaerobic conditions (5% $CO_2$, 4% $O_2$, and 91% $N_2$) produced by AnaeroGen gas packs (Oxoid). Brucella broth containing 10% (v/v) foetal bovine serum (BB10) was used for liquid H. pylori culture.

### Pull-down assays

The pull-down assays were performed as described previously[23]. Briefly, purified GST, GST-FliS and GST-FliS-HP1076 were immobilised to glutathione Sepharose (Cytiva), followed by incubating with the cell lysate of H. pylori G27 cells in the buffer containing 20 mM HEPES (pH 7.5), 137 mM NaCl, 27 mM KCl, 5% glycerol, 0.1% Tween-20 and 10 mM DTT for 3 h at 4 °C. After washing three times, the resin beads were mixed with SDS-PAGE sample loading buffer and analysed with SDS-PAGE.

### CtpA expression and purification

Fragments of DNA encoding different CtpA truncations were amplified from the cell lysate of H. pylori G27 strain and subcloned into the pAC28m vector to add an N-terminal His₆ tag. The plasmids expressing different mutants were constructed using site-directed mutagenesis based on corresponding wild-type protein expression plasmids. The primers used in this study are listed in Table 3. Proteins were expressed in E. coli BL21(DE3) after induction using 0.25 mM isopropyl-β-D-thiogalactopyranoside (IPTG) at 20 °C for 20 hours. Cells were sonicated in lysis buffer containing 20 mM Tris-HCl (pH 7.5), 150 mM NaCl and 60 mM imidazole and then centrifuged at 20,000 × g for 1 h at 4 °C to collect the supernatant. After filtering with 0.22 μm pore size filters, the supernatant was mixed and incubated with Ni-NTA agarose (Cytiva) for 1 h at 4 °C, followed by washing using the lysis buffer. Proteins were eluted with elution buffer containing 20 mM Tris-HCl (pH 7.5), 150 mM NaCl and 300 mM imidazole. The eluant was further purified with a mono S™ 4.6/100 PE column (Cytiva) in 20 mM Tris-HCl (pH 7.5), and a 150 mM to 1 M NaCl gradient and followed by gel filtration with a HiLoad® 16/600 Superdex® 200 PG column (Cytiva) in the buffer containing 20 mM Tris-HCl (pH 7.5) and 150 mM NaCl. For the purification of apo CtpA$_{TM}$, 8 M urea was added in the lysis buffer before sonication and used all through the Ni-NTA purification. After eluted with 20 mM Tris-HCl (pH 7.5), 150 mM NaCl, 60 mM imidazole and 8 M urea, the eluant was dialysed to remove urea, followed by ordinary purification steps described above.

### Activity assay of CtpA on HP1076 and immunoblot detection of HP1076

Recombinant HP1076 protein was mixed with either wild-type CtpA or CtpA mutants at a molar ratio of 10:1 in a reaction buffer containing 20 mM Tris-HCl (pH 7.5), 150 mM NaCl, and 1 mM DTT. The mixtures were incubated at room temperature for various time intervals to assess time-dependent cleavage. Reactions were terminated by the addition of SDS-PAGE loading buffer and subsequently boiled for 5 min. Protein samples

**Table 3 | Oligonucleotide sequences used in the study**

| Primer name | Sequence (5'-3') |
|---|---|
| CtpA-F | GGGAATTCCATATGTTCAGTCGTTTCTCTAATGT |
| CtpA-R | CCCAAGCTTTTATTTCTTCTTAGGCGC |
| CtpA K325A/Q329A-F | GGTGAAAAAACCTTTGGTGCGGGAAGCGTGGCGATGCTACTCCCTGTCAAT |
| CtpA K325A/Q329A-R | ATTGACAGGGAGTAGCATCGCCACGCTTCCCGCACCAAAGGTTTTTTCACC |
| CtpA H85A-F | TCTAATTTGGATGCGGCTTCAGCGTATTTGAAT |
| CtpA H85A-R | ATTCAAATACGCTGAAGCCGCATCCAAATTAGA |
| CtpA F105R-F | CAAACCGAGGGCGAAAGAGGGGGGCTTGGGATC |
| CtpA F105R-R | GATCCCAAGCCCCCCTCTTTCGCCCTCGGTTTG |
| CtpA R162A-F | GCGATCAATCTCATGGCCGGCAAGCCAAAGACC |
| CtpA R162A-R | GGTCTTTGGCTTGCCGGCCATGAGATTGATCGC |
| CtpA S300A-F | GTCAATGGCGGTTCAGCGGCCGCGAGCGAGATCGTCGCA |
| CtpA S300A-R | TGCGACGATCTCGCTCGCGGCCGCTGAACCGCCATTGAC |
| CtpA K325A-F | GGTGAAAAAACCTTTGGTGCGGGAAGCGTGCAGATGCTA |
| CtpA K325A-R | TAGCATCTGCACGCTTCCCGCACCAAAGGTTTTTTCACC |
| CtpA Q329A-F | TTTGGTAAGGGAAGCGTGGCGATGCTACTCCCTGTCAAT |
| CtpA Q329A-R | ATTGACAGGGAGTAGCATCGCCACGCTTCCCTTACCAAA |
| CtpA E382A-F | AAATTCAGCTTGAAAGCAGCGGATCTAAAACAC |
| CtpA E382A-R | GTGTTTAGATCCGCTGCTTTCAAGCTGAATTT |
| CtpA E382R-F | AAATTCAGCTTGAAACGAGCGGATCTAAAACAC |
| CtpA E382R-R | GTGTTTAGATCCGCTCGTTTCAAGCTGAATTT |

were separated by SDS-PAGE and transferred to PVDF membranes. Immunoblotting was performed using a rabbit anti-HP1076 antibody to detect both full-length HP1076 and its cleaved products. Donkey anti-rabbit IgG-HRP (Santa Cruz, Cat# sc-2313) was used as the secondary antibody.

**Cell fractionation**

An overnight liquid culture of *H. pylori* was centrifuged with $6000 \times g$ for 10 min at 4 °C to harvest the cells. The cell pellet was resuspended in 10 mM Tris-HCl (pH 8.0) to an $OD_{600}$ of 10, followed by sonication to break the cells. The cell lysate was centrifuged with 6000 g for 10 min at 4 °C to remove the cell debris and unbroken cells. The supernatant was further ultra-centrifuged with $100,000 \times g$ for 1 h at 4 °C. The supernatant containing cytosolic and periplasmic fractions and the pellet containing membrane fraction were collected for the subsequent immunoblotting analysis. Mouse anti-CtpA and rabbit anti-HP1076 antibodies were used to detect CtpA and HP1076, respectively. Mouse anti-OMP (Santa Cruz, Cat# sc-57779) was used to detect both OMP and HSP. Goat anti-mouse IgG-HRP (Santa Cruz, Cat# sc-2005) and donkey anti-rabbit IgG-HRP (Santa Cruz, Cat# sc-2313) were used as secondary antibodies to detect the corresponding primary antibodies.

**Microscale thermophoresis (MST)**

HP1076 was fluorescently labelled with the Monolith NT Protein Labelling Kit RED-NHS (Nano Temper Technologies). $CtpA_{S300A}$ was serially diluted to concentrations ranging from 165 µM to 5.04 nM, followed by mixed with 45 nM of labelled HP1076 in a 1:1 volume ratio. The assays were performed in a monolith NT.115 instrument (Nano Temper Technologies) with 70% excitation power and 40% MST power at 20 °C. The data was analysed by the MO. Affinity Analysis software.

**Protein crystallisation and structure determination**

The crystallisation experiments were set on sitting-drop vapour-diffusion crystallisation plates. The crystals of wild-type CtpA were grown in the buffer containing 100 mM bis-Tris-HCl (pH 6.5), 100 mM magnesium chloride, 25% PEG 3350 at 16 °C. The crystals of $CtpA_{F105R}$ mutant were grown in the condition A8 of Morpheus® crystallisation screen (Molecular

Dimensions). The X-ray diffraction data was collected with TPS 05 A at the National Synchrotron Radiation Research Centre, Taiwan. The datasets were processed using HKL2000[32] or iMosflm[33]. The crystals of wild-type CtpA were in the space group P12$_1$1, and the $CtpA_{F105R}$ crystals were in the space group H3. Molecular replacement was used to solve the structures with Phaser[34] in the Phenix suite[35]. Pruned and truncated structure models of Photosystem II D1 C-terminal processing protease (PDB ID: 1FC6) and CtpB from *B. subtilis* (PDB ID: 4C2E) were used together as search models to perform molecular replacement for wild-type CtpA, with six chains in the asymmetric unit. The resting protomer of wild-type CtpA was used as a search model for $CtpA_{F105R}$ mutant, with four chains in the asymmetric unit. Structure building, refinement and subsequent rebuilding were performed using COOT[36] and Phenix. The protein coordinates have been deposited to Protein Data Bank (PDB ID: 9JR1 and 9KU0). The structural images were prepared using USCF ChimeraX[37] and PyMol.

**Hydrogen deuterium exchange mass spectrometry (HDX-MS)**

$CtpA_{TM}$ in complex with HP1076 was prepared by mixing apo $CtpA_{TM}$ with HP1076 and incubated overnight at 4 °C. Stock solutions of 10 µM $CtpA_{TM}$ either in apo form or in complex with HP1076, or $CtpA_{TM/F105R}$ were prepared by diluting them in protein buffer containing 20 mM Tris-HCl (pH 7.5), 150 mM NaCl. To perform HDX experiments, 3 µl of the stock solutions were added into 47 µl of HDX buffer (20 mM Tris-HCl, 150 mM NaCl in D$_2$O, pD 7.5) at room temperature. Control experiments without HDX were performed by diluting protein samples in the corresponding deuterium-free protein buffer. After different incubation time periods (0.5, 1, 10, 60, 120 min), each HDX was quenched by adding 50 µl of pre-cooled quenching buffer (20 mM Tris-HCl (pH 1.7), 150 mM NaCl) to reach a final pH of 2.5. Immediately after quenching, 50 µl of the sample was injected into a Acquity M-Class UPLC system linked with a HDX Manager (Waters, USA) at 0 °C. The samples were online digested with an Enzymate BEH-Pepsin column (Waters, 300 Å, 5 µm, 2.1 × 30 mm) at 25 °C and subsequently desalted with a BEH C18 Trap Column (Waters, 130 Å, 5 µm, 300 × 50 mm) for a total of 3 min at 100 µl/min. The peptides were eluted through a BEH C18 column (Waters, 130 Å, 1.7 µm, 2.1 × 100 mm) using a linear gradient from 95% solvent A (H$_2$O with 0.1% formic acid) and 5%

solvent B (acetonitrile with 0.1% formic acid) to 5% solvent A and 95% solvent B over 12 minutes at a flow rate of 100 μL/min. Mass spectra were acquired with an m/z range of 100–3000 using the Waters SYNAPT G2-Si Mass Spectrometry in MS$^E$ mode. Three replicates were performed for each CtpA sample with different incubation time.

### HDX data analysis
The mass spectra raw data were processed by ProteinLynx Global Server to generate the peptide lists. The following filtering parameters were set in DynamX 3.0 to remove low-quality signal peaks: Minimum intensity: 10,000; Maximum sequence length: 30; Minimum products per amino acid: 0.3; Maximum MH+ Error (ppm): 5; Minimum PLGS score: 7. The quality of resulting signal peaks after filtering was further inspected manually. The average standard deviation of the deuterium uptake was below 0.1 Da. Difference in deuterium uptake with 95% confidence interval (CI) was considered as a significant change for differential HDX. Uptake figures were produced by Microsoft Excel and relative deuterium uptake was mapped onto the crystal model of proteins using PyMol. HX-Express3[38] was used to perform binomial fitting and bimodal deconvolution of the spectra.

### Cryo-EM sample preparation and data collection
To prepare the cryo-EM grids, 4 μl of protein sample at a concentration of 2 mg ml$^{-1}$ was applied to the glow-discharged holey carbon grids (Quantifoil R1.2/1.3). After a wait time of 5 s and a blotting time of 3 s, the grids were plunge frozen into liquid ethane using Vitrobot Mark IV (Thermo Fisher) operated at 4 °C and 100% humidity. Data acquisition was performed using a Titan Krios microscope (Thermo Fisher) operated at 300 kV and equipped with a GIF Quantum energy filter and a Gatan K3 direct electron detector. Automatic image collection was carried out using SerialEM[39] with a slit width of 20 eV on the energy filter in the super-resolution counting mode at a magnification of ×130,000 for CtpA$_{TM}$ and ×105,000 for wild-type CtpA, yielding calibrated pixel sizes of 0.92 Å and 0.83 Å, respectively. The defocus range was set from -1.5 to -2.5 μm. Each movie stack was dose-fractionated to 32 frames with a total dose of ~50 e Å$^{-2}$.

### Cryo-EM data processing and map calculation
The movies collected were motion-corrected with MotionCor2[40], followed by contrast transfer function estimation (CTF) with CTFFIND[41]. Image processing was performed using Relion[42] and cryoSPARC software[43]. Particles were auto-picked and processed with 2D classification using Relion. Noise and junk particles were removed by multiple rounds of 2D classification. Selected 2D classes were used for automated particle picking by Gautomatch (http://www.mrc-lmb.cam.ac.uk/kzhang/). After removing junk particles by multiple rounds of 2D classification, ab initio reconstruction and heterogeneous refinement were performed to identify the best classes of particles. Non-uniform refinement was further applied to generate the final cryo-EM density maps.

For wild-type CtpA, 1,583,803 particles were picked by template picking and further extracted and imported into cryoSPARC. The particles were subjected to 2D classification with a mask diameter of 226 Å. 561,739 particles were left to perform ab initio reconstruction. After heterogeneous refinement in cryoSPARC, 155,703 particles were selected and subjected to nonuniform refinement with C3 symmetry followed by local refinement, yielding a map for CtpA conformation I with an overall resolution of 3.4 Å (EMD-62575); 133,262 particles were selected and refined to yield a map for CtpA conformation II with an overall resolution of 3.6 Å (EMD-62576); 144,375 particles were selected and refined to yield a map for CtpA conformation III with an overall resolution of 3.5 Å (EMD-62553).

For CtpA$_{TM}$, 1,449,887 particles were picked by template picking and further extracted and imported into cryoSPARC. The particles were subjected to 2D classification with a mask diameter of 226 Å. 381,365 particles were left to perform ab initio reconstruction. After heterogeneous refinement in cryoSPARC, 151,917 particles were selected and subjected to nonuniform refinement with C3 symmetry followed by local refinement, yielding a map with an overall resolution of 3.1 Å (EMD-62573).

### Model building and coordinate refinement
An initial predicted model was generated with DeepTracer[44–46] after submitting the cryo-EM map of CtpA$_{TM}$ and its sequence file in fasta format. The crystal structure of wild-type CtpA was superimposed onto the initial model and modified to fit the density in COOT. The missing structures were manually built in COOT. Cycles of model building and refinement were performed using COOT and real_space_refine in Phenix. The geometry of the model was refined with the ISOLDE[47] plugin in ChimeraX. The final refined model was validated using MolProbity[48]. The protein coordinates have been deposited to Protein Data Bank (PDB ID: 9KU3, 9KUB, 9KUC and 9KSP) The hydrogen bonds and contact interfaces were analysed with the PDBePISA server[49]. The structural images were prepared using USCF ChimeraX and PyMol.

### Absorbance-based analytical ultracentrifugation
Sedimentation velocity studies of CtpA were performed using a Beckman proteomeLab XL-I analytical ultracentrifuge equipped with an An-60 Ti rotor at 16 °C. Double-sector centrifuge cells were loaded with 380 μl of the sample and 400 μl of the reference buffer. Data from sedimentation at 28,000 rpm were collected at 4 min intervals for 12 h in a continuous mode. Sedimentation velocity data were fitted to a c(s) continuous size distribution model using SEDFIT[50] to determine the sedimentation coefficients.

### Processive proteolysis monitored by reverse-phase HPLC
HP1076 (50 μg) was mixed with CtpA (5 μg) in the buffer containing 20 mM Tris-HCl (pH 7.5) and 150 mM NaCl and incubated for 0, 15, 30, 60, 90 min at room temperature. The reaction was quenched by adding the same volume of 7.4 M guanidine hydrochloride. For the negative control, HP1076 (50 μg) was mixed with trypsin (3 μg) and incubated for 0, 15, 30, 60, 90 min at 37 °C, followed by quenching of the reaction by adding the same volume of 7.4 M guanidine hydrochloride. Immediately after quenching, 5 μl of the sample was injected onto an XBridge BEH C18 column (Waters, 130 Å, 5 μm, 4.6 × 250 mm) at a flow rate of 1.2 ml/min with a 1260 Infinity II LC System equipped with a 1290 Infinity II Diode Array Detector (Agilent). A 0 to 100% acetonitrile in 0.1% trifluoroacetic acid gradient in 30 min was performed to elute the degradation products at a flow rate of 1.2 ml/min. The peaks were detected at 210 nm wavelength.

### Statistics and reproducibility
For HDX-MS analysis, mass differences greater than 0.19 Da were considered significant. Statistical comparisons between CtpA$_{TM}$ with and without HP1076 binding were performed using Student's t-test, with significance defined as $p < 0.05$. Each dataset was generated from more than three independent experiments. All other experiments were also independently repeated at least three times to ensure reproducibility.

### Reporting summary
Further information on research design is available in the Nature Portfolio Reporting Summary linked to this article.

### Data availability
The cryo-EM maps and atomic coordinates of CtpA and CtpA$_{TM}$ have been deposited in the Electron Microscopy Data Bank (https://www.ebi.ac.uk/pdbe/emdb/) under accession number EMD-62576, EMD-62553, EMD-62573, and EMD-62575, and in the Protein Data Bank (https://www.rcsb.org) under accession number 9KUC, 9KSP, 9KU3, and 9KUB, respectively. The crystal structure models of CtpA and CtpA$_{F105R}$ have been deposited in the Protein Data Bank under accession number 9JR1 and 9KU0, respectively. The HDX-MS data have been deposited to the ProteomeXchange Consortium via the PRIDE[51] partner repository with the dataset identifier PXD068881. The numerical source data underlying the graphs presented in this study are available in Supplementary Data 1 and 2. Uncropped and unedited blot and gel images are provided in Supplementary Fig. 15. All other data are available from the corresponding author upon reasonable request.

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

## Acknowledgements
This research was funded by the Hong Kong Research Grants Council (General Research Fund 14117622 and Collaborative Research Fund C4012-16E) and National Natural Science Foundation of China (NSFC, No. 31900046 and 82372269 to H.Z., No. 82172465 to D.W.). We extend our gratitude to the staff at beamlines BL15A and TPS05A of the National Synchrotron Radiation Research Centre, Taiwan, China. The authors thank the Cryo-EM Centre of Southern University of Science and Technology for data collection and HPC-Service Station.

## Author contributions
Conceptualization, K.S. and S.W.N.A.; methodology, K.S., P.K.S., S.P.K.T., H.Z., and S.W.N.A.; investigation, K.S., L.Y., C.Y.M., P.K.S., S.P.K.T., and H.Z.; writing – original draft, K.S.; writing – review and editing, K.S., P.K.S., S.P.K.T, H.Z., and S.W.N.A.; funding acquisition, S.W.N.A.; resources, K.F.L., D.W., H.Z., and S.W.N.A.; supervision, S.W.N.A.

## Competing interests
The authors declare no competing interests.
