## [Transparent Peer Review file · Communications Biology]

Structures of *Helicobacter pylori* C-terminal protease CtpA reveal a new mode of self-contained proteolytic processing

Corresponding Author: Professor Shannon Wing Ngor Au

Version 0:

Reviewer comments:

Reviewer #1

(Remarks to the Author)

The manuscript by Sun and colleagues reports on biophysical and structural studies by cryo-EM of the carboxyl-processing peptidase CtpA from the enteropathogen *Helicobacter pylori* (HpCtpA). CtpA enzymes have been characterized from several bacteria and display diversity in their structural organization. Here, the authors characterize the activation and oligomerization of the HpCtpA form. The study is very interesting and suitable for Communications Biology. However, the following issues should be adequately addressed prior to definitive acceptance of the manuscript.

L84: And the crystal structures?

L97: Any further information about HP1076? From PMID 20581225 from the same group it seems to be a co-chaperone.

L105-106: The authors should show—not only suggest—that HP1076 is cleaved. Isn't this derivable from Suppl. Fig. 1 A and B?

Fig. 1A: Please, include the limiting residues of each domain/segment.

Suppl. Table 1:

(1) Given the resolution of the data, please limit the significant digits in the cell constants to one decimal.

(2) Other values (completeness, B-factors, etc.) should also be presented with less precision.

(3) In contrast, the rmsd values of bond lengths and angles should include a third decimal.

(4) Please, indicate in the legend what the three Ramachandran-plot values stand for.

(5) Is there an explanation for the substantially worse refinement parameters of the mutant compared to the wild type? The difference in resolution is merely 0.3 Å.

Suppl. Table 2: Is the clashscore of EMD-62573 really 0.11?

Suppl. Tables 4 and 5: Given the resolution of the structures, can all these interactions be assigned confidently?

The manuscript would benefit from a thorough linguistic revision.

Some—not comprehensive—examples:

Line 34 (L34): ...binding. Cryo-electron microscopy and X-ray crystallography structures....

L53: Although the orientation....

L54: ...unit is structurally conserved.

L65, L66, L70, etc.

Reviewer #2

(Remarks to the Author)

Structures of *Helicobacter pylori* C-terminal protease CtpA reveal a new mode of self-contained proteolytic processing

The experimental work by Sun et al sheds light on the *H. pylori* carboxy-processing protease (CtpA), an essential protein in bacteria for the processing and degradation of proteins. Many questions remain on how these proteases become active and their conformational changes in response to substrates. Using a suite of biophysical techniques (AUC, X-ray crystallography, Cryo-EM, in vitro proteolysis assays and hydrogen-deuterium exchange mass spectrometry), they show the oligomeric state (trimer of dimers) of *H. pylori* CtpA and how only one monomer of each dimer can be in the active state to allow recognition and processing of substrates. Overall, the work is of high quality and reveals the molecular details of

activation of *H.pylori* CtpA through very informative text and figures.

One potential problem arises with the analysis of the HDX-MS data concerning apo CtpATM protein. As it exists in at least conformations (according to the Cryo-EM data), it is very difficult to assess what is protected and what is not due to potential interconversion of the subunits between the active and inactive state. Sun et al. do surmise this on lines 154-155 and reason that the resting dominates the population. Do you have any HDX-MS data on the CtpATM F105R where the hexamer is entirely in the resting state and are there any differences with apo CtpATM.

Recommend minor edits.

1. Line 48 – The CTPs are categorized into the three subfamilies. It is not explicitly stated which HpCtpA is (I am assuming CTP-3).

2. Lines 51-53 - ...consists of an N-terminal domain, a protease domain which can be further divided into a cap subdomain and a core subdomain, a PDZ domain, and a C-terminal domain

Should be

...consists of an N-terminal domain, a PDZ domain, a protease domain which can be further divided into a cap subdomain and a core subdomain, and a C-terminal domain

3. Lines 53-56

Despite the orientation of the substrate-binding PDZ domain is highly flexible, the PDZ-protease unit is the most structurally conserved. On the contrary, the N- and C-terminal domains participating in intermolecular interactions that generate different CTP oligomeric forms are variable.

Rewrite and combine the sentences to something like

Despite the highly flexible orientation of the substrate-binding PDZ domain, the PDZ-protease unit is the most structurally conserved whereas the N- and C-terminal domains participating in intermolecular interactions that generate different CTP oligomeric forms are variable.

4. Lines 86-87

H. pylori is the primary pathogen colonizing human stomachs and highly related to gastric cancer (17)

Change to something like

H. pylori is the primary pathogen colonizing human stomachs and its presence strongly correlates with the incidence of gastric cancer (17).

5. Line 96-97.

Can you briefly explain what HP1076 is.

6. Lines 107-108

Analytic ultracentrifugation revealed CtpA is a hexamer in solution (Figure S1E) and presented as a triangular ring structure (Figure S1F).

Changed to something like

Analytic ultracentrifugation revealed CtpA is a hexamer in solution (Figure S1E) and presented as a triangular ring structure as revealed by TEM (Figure S1F).

7. Lines 124-125

Each face of CtpATM hexamer carries three subunits at the same state, either active or resting (Figure 1BC). For the face with active subunits, ...

Change to something like

Each face of CtpATM hexamer is comprised of three subunits in the same state, either active or resting (Figure 1BC). For the active face, ...

Or

Each face of CtpATM hexamer is comprised of three subunits in the same state, either active or resting (Figure 1BC). For the active face side of the hexamer, ...

8. Lines 135-136

Specifically, ML either contacts with the PDZ domain or inserts into the cleft between an active and a resting subunit.

It should either be ...contacts the PDZ domain... or ...interacts with the PDZ domain...

9. Lines 144-146

Construct CtpATM was used and its apo form was purified with denaturation in urea and refolding to remove the bound endogenous substrate from *E. coli*.

Change to something like

The CtpATM construct was denatured with urea, refolded and purified to remove the endogenously bound substrate from *E. coli* (apo CtpTM).

10. Lines 159-161

Next, to monitor the molecular dynamics of CtpA during substrate binding, excessive HP1076 was added into the apo CtpATM sample and incubated to retain CtpATM to the active conformation to the most extent.

I do not understand this sentence. Are you incubating with an excess of HP1076 to try and shift the equilibrium of the hexamer subunits towards the active state?

11. Lines 172-177

The bimodal spectra distribution of $\alpha 3$ and the ML region was also found in CtpA-HP1076 sample (Figure S7), indicating that when CtpA was saturated by the substrate, the structure still contained a combination of subunits with different conformations. This is consistent with the crystal and cryo-EM structures obtained that half of subunits in one hexamer are at the resting state.

And the addition of a large excess of substrate did not shift the equilibrium to the active state?

12. Lines 185-186

...corresponding to CtpA at different conformations:

Change to

...corresponding to CtpA in different conformations

13. Lines 196-209

There are a lot of important details in this section. It may be better to split some of the longer sentences into two. For example.

A detailed structural analysis revealed that extensive hydrogen-bonds and salt bridges are formed among the two NTDs and the two protease cap subdomains (excluding $\alpha 3$) of the two subunits in one N-dimer, therefore grouping these parts into a rigid and isolated module (referred to as the dynamic unit) (Figure 3B and Table S4).

to

A detailed structural analysis revealed that extensive hydrogen-bonds and salt bridges are formed between the two NTDs and protease cap subdomains (excluding $\alpha 3$) of each subunit in a single N-dimer. This group of interactions form a rigid and isolated module (referred to as the dynamic unit) (Figure 3B and Table S4).

You can remove "on the other hand"

14. Line 212.

You have described on line 191 NTD (N-dimer), but you have not described C-dimer.

15. Lines 212-213

...C-dimer partners in CtpATM and CtpA conformation I and conformation II.

Change to

...C-dimer partners in both CtpATM and CtpA in conformation I and conformation II.

16. Line 271. R162E. In Supplementary figure 8, it is labelled as R162A. Which is correct?

17. Line 296 ...HpCtpA with only one subunit... should be ...HpCtpA where only one subunit...

18. Lines 298-300

The concerted movement of the PDZ domain and the motile loops from the C-dimer partners during the conversion between resting and active states warrant only unfolded C-terminus of the substrate enters to the active subunit and be processed in a processive manner

Should be...

The concerted movement of the PDZ domain and the motile loops from the C-dimer partners during the conversion between resting and active states only allow unfolded C-termini of substrates to enter the active subunit and be processed in a processive manner.

Or

The concerted movement of the PDZ domain and the motile loops from the C-dimer partners during the conversion between resting and active states only allow the unfolded C-terminus of a substrate to enter the active subunit and be processed in a processive manner.

19. Furthermore, Lines 300-301

This work also explains the role of conserved Arg-162 and Phe-105 in interacting with the terminus of substrate in the activation process and opens a new perspective to understand the structural basis of C-terminal proteases.

Depend on which you choose for Lines 298-300, it should either be

This work also explains the role of conserved Arg-162 and Phe-105 in interacting with the C-terminus of a substrate in the activation process and opens a new perspective to understand the structural basis of C-terminal proteases.

Or

This work also explains the role of conserved Arg-162 and Phe-105 in interacting with the C-termini of substrates in the activation process and opens a new perspective to understand the structural basis of C-terminal proteases.

20. Line 313 – After filtered with 0.22um... After filtering with a 0.22um...

Reviewer #3

(Remarks to the Author)

In the manuscript entitled "Structures of *Helicobacter pylori* C-terminal protease CtpA reveal a new mode of self-contained proteolytic processing" by Sun et al., the authors have used structural and hydrogen-deuterium exchange mass spectrometry (HDX-MS) techniques to understand the structural basis of bacterial Carboxyl-terminal processing proteases (CTP's) using *Helicobacter pylori* CTP, HpCtpA. Using the above-mentioned techniques, the authors have demonstrated the probable

conformational transitions involved in transition of protein from resting (inactive) to active states. The manuscript seems exciting given the conformational variability, presence of adaptor proteins and substrates involved in the functionality of CTP's, known so far. However, there is a major concern in the way the manuscript is written, and the results are demonstrated, it seems that representation of data is not justified. Given the data in hand and the scope of this journal, the authors should consider restructuring the manuscript and data presentation for better understanding to the readers from the field as well as outside the field. The following are the major concerns for this manuscript:

Introduction:

Major Concern: The introduction seems missing to demonstrate the major significance of CTP's as major targets of bacterial infections. The authors should consider re-writing the introduction to better address the importance of CTP's in terms of human health and how these proteins can be used as targets against bacterial associated diseases. Moreover, majority of the introduction (lines 51-83) seems to be describing the structural features between the HpCtpA homologs structurally characterized in other species. This section seems to be more appropriate in the discussion section of the manuscript rather than the introduction. Focusing the introduction on the need to study CTP's, differences categories of this family of proteins, major highlights of the research done on these enzymes, their implications and the need for more studies on similar enzymes will help the reader to better understand the importance of these enzymes in terms of human health.

Minor concerns:

1. The sequence of a typical CTP and its description in the text is not properly mentioned, moreover, the authors should pay particular attention to the use of acronyms throughout the text. Since a lot of acronyms are used throughout the manuscript the authors should properly introduce them when they are first used.
2. A schematic figure describing a typical CTP might be helpful to properly describe the enzyme, its domains and the movement during functioning of the enzyme might be helpful.
3. The authors can also elaborate on the importance of HpCtpA (lines 80-92) to give the reader a better understanding of the need and scope of present study.

Results and Discussion:

Major concerns: The authors should consider major changes in this section of the manuscript. The following are the major concerns:

Structures of *H. pylori* CtpA:

1. The authors never described anything about HP1076 protein? It's cellular location? What is the molecular weight of full protein and its degraded forms? Is it an adaptor protein for HpCtpA or the substrate? Also based on the CtpA's classification HpCtp belongs to CtpA or CtpB class? The authors should clarify on this.
2. How do the authors detect and validate the presence of HpCtpA and degraded HP1076 in the pull-down assay? The methods to describe how pull-down experiments were carried out is not mentioned in M&M.
3. Ultracentrifugation results also show that another population of 474 kD is observed, authors should explain this.
4. The authors have mentioned that HpCtpA recombinant construct lacks first 20aa, which are probably signal peptide but didn't mention the function of signal peptide? The rationale for making CtpA Δ N46 mutant which they are using throughout this study. Why is CtpAS300A an inactive mutant? Why α - is mentioned against the names of each protein in Fig S1D? Based on WB in Fig S1D, both protein HpCtpA and HP1076 seem to be present both in membrane and cytosolic fractions, what is the explanation for these observations?
5. It seems like MST is used to derive the Kd between the inactive CtpAS300A and HP1076, why is active protein not used for this experiment? Also MST experiment doesn't seem to reach the saturation limit of the substrate, moreover error bars are missing for the experiment.
6. What is the rationale of using the term "resting state" rather than inactive state as used in the field?
7. The authors claim that "Based on the orientations of the PDZ domains and the presence of substrate peptides, the CtpA crystal structure contains three subunits in the resting state, two subunits in the active state and one protomer in an intermediate state". To better demonstrate this observation, authors should define either a resting/active state monomer and then demonstrate a side-by-side comparison of the different states. This figure should be incorporated in the main figures of the paper. This will give a better understanding of the structural features of the protein and demonstrate the intramolecular movements in different states as authors have shown in supp. Figure 2 B-E. Also the figure legend for Supply Figure 2 is hard to understand.
8. The authors didn't provide the rationale for designing CtpAS300A/K325A/Q329A, CtpATM mutant. This mutant was supposed to be catalytically inactive, but the cryoEM map shows that the hexameric structure still contains both active and resting states, this observation should be explained.
9. For CtpATM mutant the authors mentioned that "For the face with active subunits, a substrate peptide modeled with poly-alanine was assigned next to their shifted PDZ domain". The authors should provide a figure for this statement, also if the active state in the hexamer is substrate induced that means the inherent tendency of HpCtpA is to remain in resting state?
10. The authors mentioned the structural regions involved in dimerization and those associated with transition from resting to active states, since similar movements were observed in BsCtpB and PaCtpA, the authors should elaborately compare present structure with the published structures and mention the similarities and differences between these structures.
11. Figure citing the movement ML loop movement is missing.

Structural dynamics of CtpA

1. Doesn't the statement "It is plausible that CtpA subunits are in an equilibrium between the active and resting states in solution independent of substrate binding" classify HpCtpA to CtpB subfamily?
Asymmetric conformational changes and the assembly of self-compartmentalization unit in CtpA
1. The CryoEM structure was mentioned in the beginning of results but elaborately explained here. The authors should

consider explaining the structural details in one heading to readers will have a holistic understanding of HpCtpA structure.
2. Based on structural data analysis, the authors provided plausible explanations of the presence of different states of protomers and functioning of different domains/residues in the functioning of this protein but the detailed figures explaining this text is missing. The authors should consider making figures explaining the text in lines 200-209, 218-224 and 227-233 to provide a better understanding of how this protein works.

Minor Comments:

1. The authors a lot of acronyms in the results and discussion sections also, such as N-dimer, subunit Sa, Sr, ML etc. kindly explain while explaining the structure probably in the beginning where structural introduction is provided for better readability throughout the manuscript.
2. Some of the figures are hard to understand, for example Supply Fig S2, it is hard to follow the figure captions.
3. In some figures, especially structural figures, it is hard to see the text as it is embedded in the figure, kindly make necessary changes for better visualization.
4. There is a lot of text where citations are missing, kindly include the appropriate references.

Version 1:

Reviewer comments:

Reviewer #1

(Remarks to the Author)

The authors have addressed all my concerns adequately except the most important: they must show that that HP1076 is cleaved. Without this simple experiment, I cannot accept the manuscript.

Reviewer #2

(Remarks to the Author)

I wish to thank the authors for addressing my questions/recommendations/concerns as well as those by the other reviewers.

The manuscript adds to the scientific knowledge of bacterial C-terminal proteases and should be accepted for publication as it now stands.

Reviewer #3

(Remarks to the Author)

The revised version provides a clearer presentation and better overall understanding of the data. However, a few minor issues remain that should be addressed:

Introduction referencing: The introduction still appears under-referenced. For example, lines 68–73 discuss the architecture of CTPs, but relevant references are missing.

Figure citation alignment:

Lines 68–70 correspond to Figure 1A but this is not cited in the text.

Lines 77–82 correspond to Figure 1B, but this figure is also not cited.

Protein truncation screening: The authors should explicitly mention that they screened multiple N-terminal truncated versions of the protein and selected CtpA Δ N46 as the most stable construct retaining the longest N-terminal end.

Catalytic triad referencing: Since Serine 300 belongs to the well-established catalytic triad of CTPs, appropriate references to previous studies should be included whenever these residues are discussed.

HP1076 localization: I disagree with the authors' statement that the HP1076 band in the C/P fraction is "very weak." Based on the data, the authors can state that HP1076 is predominantly present in the membrane fraction. However, the data does not support the stronger conclusion that this protein is only membrane-localized.

Figure 4B clarity: In the enlarged view of Figure 4B, the interactions and residues in the dynamic unit are difficult to discern. Making the ribbons transparent would help highlight these interactions more clearly.

Structural comparison (lines 229–237): As suggested by the text, a supplementary figure comparing the structures of BsCtpB, HpCtpA, and PaCtpA would be very valuable to illustrate how the C-terminal region determines the oligomerization state of CTPs.

Figure 4C inset and Supplementary Figure S9: The interactions and residues in the inset of Figure 4C are also difficult to visualize. It would be very helpful if the important residues mentioned in the text could be clearly highlighted in Supplementary Figure S9.

There are still several places in the manuscript where figures are not cited adequately. These should be carefully cross-checked and corrected.

Reviewer #1 (Remarks to the Author):

The manuscript by Sun and colleagues reports on biophysical and structural studies by cryo-EM of the carboxyl-processing peptidase CtpA from the enteropathogen Helicobacter pylori (HpCtpA). CtpA enzymes have been characterized from several bacteria and display diversity in their structural organization. Here, the authors characterize the activation and oligomerization of the HpCtpA form. The study is very interesting and suitable for Communications Biology. However, the following issues should be adequately addressed prior to definitive acceptance of the manuscript.

>> We thank the reviewer for this positive and constructive feedback. Please see below our response to address the comments point by point.

L84: And the crystal structures?

>> We have revised the statement and included “protein crystallography” (see line 92).

L97: Any further information about HP1076? From PMID 20581225 from the same group it seems to be a co-chaperone.

>> We have included some information about HP1076 as suggested (see line 99-101).

L105-106: The authors should show—not only suggest—that HP1076 is cleaved. Isn't this derivable from Suppl. Fig. 1 A and B?

>> We have revised the statement accordingly (see line 121).

Fig. 1A: Please, include the limiting residues of each domain/segment.

>> We have revised Figure 1A and included the residue range of each domain.

Suppl. Table 1:

(1) Given the resolution of the data, please limit the significant digits in the cell constants to one decimal.

(2) Other values (completeness, B-factors, etc.) should also be presented with less precision.

(3) In contrast, the rmsd values of bond lengths and angles should include a third decimal.

(4) Please, indicate in the legend what the three Ramachandran-plot values stand for.

(5) Is there an explanation for the substantially worse refinement parameters of the mutant compared to the wild type? The difference in resolution is merely 0.3 Å.

>> We have revised the table contents to the suggested decimal place (see Suppl. Table 1). For the refinement of CtpA mutant F105A, we consider the worse refinement statistics

is due to the weak electron density map of the hinge region of the PDZ domain, PDZ domain and residues close to the motile loop. These regions have lower correlation coefficient (CC) when compared with the equivalent regions in the wild-type CtpA crystal structure.

Suppl. Table 2: Is the clashscore of EMD-62573 really 0.11?

>> Yes, the clashscore was generated from the last cycle of structure refinement in Phenix.

Suppl. Tables 4 and 5: Given the resolution of the structures, can all these interactions be assigned confidently?

>> These interactions were identified by PISA. We have updated the list in Tables 4 and 5 and only included interactions within 3.2 angstrom.

The manuscript would benefit from a thorough linguistic revision.

>> Our revised manuscript has been proofread by professional academic editing.

Some-not comprehensive-examples:

Line 34 (L34): ...binding. Cryo-electron microscopy and X-ray crystallography structures....

>> We revised the statement to “Cryo-electron microscopy and crystal structural analysis of CtpA further...” (see line 32 and 33).

L53: Although the orientation....

>> As also suggested by Reviewer #2, we have revised the statement (see line 70-73).

L54: ...unit is structurally conserved.

>> We have revised the statement (see line 71).

L65, L66, L70, etc.

>> Our manuscript has undergone professional academic editing and comprehensive linguistic revision.

Reviewer #2 (Remarks to the Author):

The experimental work by Sun et al sheds light on the *H. pylori* carboxy-processing protease (CtpA), an essential protein in bacteria for the processing and degradation of proteins. Many questions remain on how these proteases become active and their conformational changes in response to substrates. Using a suite of biophysical techniques (AUC, X-ray crystallography, Cryo-EM, in vitro proteolysis assays and hydrogen-deuterium exchange mass spectrometry), they show the oligomeric state (trimer of dimers) of *H. pylori* CtpA and how only one monomer of each dimer can be in the active state to allow recognition and processing of substrates. Overall, the work is of high quality and reveals the molecular details of activation of *H. pylori* CtpA through very informative text and figures.

>> We thank the reviewer for these positive comments and constructive suggestions. Please find our response to the feedback below.

One potential problem arises with the analysis of the HDX-MS data concerning apo CtpATM protein. As it exists in at least conformations (according to the Cryo-EM data), it is very difficult to assess what is protected and what is not due to potential interconversion of the subunits between the active and inactive state. Sun et al. do surmise this on lines 154-155 and reason that the resting dominates the population. Do you have any HDX-MS data on the CtpATM F105R where the hexamer is entirely in the resting state and are there any differences with apo CtpATM.

>> We have carried out HDX-MS on CtpA_{TM/F105R} and the data showed that F105R mutation impaired the conformational transition from the resting to the active state (see line 307-311 and Supplemental S12). Bimodal pattern of peptides covering residues 90–113 and 394–422 were not observed in CtpA_{TM/F105R} mutant.

Recommend minor edits.

1. Line 48 – The CTPs are categorized into the three subfamilies. It is not explicitly stated which HpCtpA is (I am assuming CTP-3).

>> HpCtpA belongs to CTP-3 subfamily. We revised the statement (see line 94).

2. Lines 51-53 - ...consists of an N-terminal domain, a protease domain which can be further divided into a cap subdomain and a core subdomain, a PDZ domain, and a C-terminal domain

Should be

...consists of an N-terminal domain, a PDZ domain, a protease domain which can be further divided into a cap subdomain and a core subdomain, and a C-terminal domain

>> We have revised as suggested (see line 68-70).

3. Lines 53-56

Despite the orientation of the substrate-binding PDZ domain is highly flexible, the PDZ-protease unit is the most structurally conserved. On the contrary, the N- and C-terminal domains participating in intermolecular interactions that generate different CTP oligomeric forms are variable.

Rewrite and combine the sentences to something like

Despite the highly flexible orientation of the substrate-binding PDZ domain, the PDZ-protease unit is the most structurally conserved whereas the N- and C-terminal domains participating in intermolecular interactions that generate different CTP oligomeric forms are variable.

>> We have revised as suggested (see line 70-73).

4. Lines 86-87

H. pylori is the primary pathogen colonizing human stomachs and highly related to gastric cancer (17)

Change to something like

H. pylori is the primary pathogen colonizing human stomachs and its presence strongly correlates with the incidence of gastric cancer (17).

>> We have revised as suggested (see line 94-96).

5. Line 96-97.

Can you briefly explain what HP1076 is.

>> We have included some information about HP1076 (see line 99-101)

6. Lines 107-108

Analytic ultracentrifugation revealed CtpA is a hexamer in solution (Figure S1E) and presented as a triangular ring structure (Figure S1F).

Changed to something like

Analytic ultracentrifugation revealed CtpA is a hexamer in solution (Figure S1E) and presented as a triangular ring structure as revealed by TEM (Figure S1F).

>> We have revised as suggested (see line 123-125).

7. Lines 124-125

Each face of CtpATM hexamer carries three subunits at the same state, either active or resting (Figure 1BC). For the face with active subunits, ...

Change to something like

Each face of CtpATM hexamer is comprised of three subunits in the same state, either active or resting (Figure 1BC). For the active face, ...

Or

Each face of CtpATM hexamer is comprised of three subunits in the same state, either active or resting (Figure 1BC). For the active face side of the hexamer, ...

>> We have revised the statement to “For the active face side of the hexamer,...” as suggested (see line 144-145).

8. Lines 135-136

Specifically, ML either contacts with the PDZ domain or inserts into the cleft between an active and a resting subunit.

It should either be ...contacts the PDZ domain... or ...interacts with the PDZ domain...

>> We have revised the statement to “.. contacts the PDZ domain” as suggested (see line 155).

9. Lines 144-146

Construct CtpATM was used and its apo form was purified with denaturation in urea and refolding to remove the bound endogenous substrate from E. coli.

Change to something like

The CtpATM construct was denatured with urea, refolded and purified to remove the endogenously bound substrate from E coli (apo CtpTM).

>> We have revised as suggested (see line 164-165).

10. Lines 159-161

Next, to monitor the molecular dynamics of CtpA during substrate binding, excessive HP1076 was added into the apo CtpATM sample and incubated to retain CtpATM to the active conformation to the most extent.

I do not understand this sentence. Are you incubating with an excess of HP1076 to try and shift the equilibrium of the hexamer subunits towards the active state?

>> Yes, the incubation with excess of HP1076 aimed to shift the equilibrium of the hexamer subunits towards the active state. We have revised the statement to clarify the message (see line 178-181).

11. Lines 172-177

The bimodal spectra distribution of $\alpha 3$ and the ML region was also found in CtpA-HP1076 sample (Figure S7), indicating that when CtpA was saturated by the substrate, the structure still contained a combination of subunits with different conformations. This is consistent with the crystal and cryo-EM structures obtained that half of subunits in one hexamer are at the resting state.

And the addition of a large excess of substrate did not shift the equilibrium to the active state?

>> **Yes. As described in next section of our manuscript, we identified CtpA underwent asymmetric conformational changes that prevented all six subunits from shifting the equilibrium to the active state. We have included a statement to clarify the message (see line 197 and 198).**

12. Lines 185-186

...corresponding to CtpA at different conformations:

Change to

...corresponding to CtpA in different conformations

>> >> **We have revised the statement as suggested (see line 206 and 207).**

13. Lines 196-209

There are a lot of important details in this section. It may be better to split some of the longer sentences into two. For example.

A detailed structural analysis revealed that extensive hydrogen-bonds and salt bridges are formed among the two NTDs and the two protease cap subdomains (excluding $\alpha 3$) of the two subunits in one N-dimer, therefore grouping these parts into a rigid and isolated module (referred to as the dynamic unit) (Figure 3B and Table S4).

to

A detailed structural analysis revealed that extensive hydrogen-bonds and salt bridges are formed between the two NTDs and protease cap subdomains (excluding $\alpha 3$) of each subunit in a single N-dimer. This group of interactions form a rigid and isolated module (referred to as the dynamic unit) (Figure 3B and Table S4).

You can remove “on the other hand”

>> **We have revised the statement as suggested (see line 217-220).**

14. Line 212.

You have described on line 191 NTD (N-dimer), but you have not described C-dimer.

>> **We have revised and described the C-dimer in line 240-241.**

15. Lines 212-213

...C-dimer partners in CtpATM and CtpA conformation I and conformation II.

Change to

...C-dimer partners in both CtpATM and CtpA in conformation I and conformation II.

>> We have revised as suggested (see line 241).

16. Line 271. R162E. In Supplementary figure 8, it is labelled as R162A. Which is correct?

>> In Line 271 of the original manuscript, that should be R162A. After receiving the comments of the reviewer, we further examined the activity of R162E. As shown in the revised manuscript, Supplemental figure S8, R162E exhibited a more severe inhibitory effect on enzyme activity. We have revised the results accordingly (see line 298-299).

17. Line 296 *...HpCtpA with only one subunit... should be ...HpCtpA where only one subunit...*

>> We have revised the text as suggested (see line 328).

18. Lines 298-300

The concerted movement of the PDZ domain and the motile loops from the C-dimer partners during the conversion between resting and active states warrant only unfolded C-terminus of the substrate enters to the active subunit and be processed in a processive manner

Should be...

The concerted movement of the PDZ domain and the motile loops from the C-dimer partners during the conversion between resting and active states only allow unfolded C-termini of substrates to enter the active subunit and be processed in a processive manner.

Or

The concerted movement of the PDZ domain and the motile loops from the C-dimer partners during the conversion between resting and active states only allow the unfolded C-terminus of a substrate to enter the active subunit and be processed in a processive manner.

>> We have revised the text as suggested (see line 329-332).

19. Furthermore, Lines 300-301

This work also explains the role of conserved Arg-162 and Phe-105 in interacting with the terminus of substrate in the activation process and opens a new perspective to understand the structural basis of C-terminal proteases.

Depend on which you choose for Lines 298-300, it should either be

This work also explains the role of conserved Arg-162 and Phe-105 in interacting with the C-terminus of a substrate in the activation process and opens a new perspective to understand the structural basis of C-terminal proteases.

Or

This work also explains the role of conserved Arg-162 and Phe-105 in interacting with the C-termini of substrates in the activation process and opens a new perspective to understand the structural basis of C-terminal proteases.

>> We have revised the text as suggested (see line 332-334).

20. Line 313 – After filtered with 0.22um... After filtering with a 0.22um...

>> We have revised the text as suggested (see line 354).

Reviewer #3 (Remarks to the Author):

In the manuscript entitled “Structures of Helicobacter pylori C-terminal protease CtpA reveal a new mode of self-contained proteolytic processing” by Sun et al., the authors have used structural and hydrogen-deuterium exchange mass spectrometry (HDX-MS) techniques to understand the structural basis of bacterial Carboxyl-terminal processing proteases (CTP’s) using Helicobacter pylori CTP, HpCtpA. Using the above-mentioned techniques, the authors have demonstrated the probable conformational transitions involved in transition of protein from resting (inactive) to active states. The manuscript seems exciting given the conformational variability, presence of adaptor proteins and substrates involved in the functionality of CTP’s, known so far. However, there is a major concern in the way the manuscript is written, and the results are demonstrated, it seems that representation of data is not justified. Given the data in hand and the scope of this journal, the authors should consider restructuring the manuscript and data presentation for better understanding to the readers from the field as well as outside the field. The following are the major concerns for this manuscript:

>> We thank the reviewer for the constructive suggestions. Please refer to our response to the feedback below.

Introduction:

Major Concern: The introduction seems missing to demonstrate the major significance of CTP’s as major targets of bacterial infections. The authors should consider re-writing the introduction to better address the importance of CTP’s in terms of human health and how these proteins can be used as targets against bacterial associated diseases. Moreover, majority of the introduction (lines 51-83) seems to be describing the structural features between the HpCtpA homologs structurally characterized in other species. This section seems to be more appropriate in the discussion section of the manuscript rather than the introduction. Focusing the introduction on the need to study CTP’s, differences categories of this family of proteins, major highlights of the research done on these enzymes, their implications and the need for more studies on similar enzymes will help the reader to better understand the importance of these enzymes in terms of human health.

>> We have substantially revised the introduction to better highlight the importance of CTPs in bacterial physiology and pathogenesis, with a clearer emphasis on their potential as therapeutic targets. Specifically, we now discuss the roles of CTPs in peptidoglycan remodelling, protein quality control, and stress response, as well as their relevance to virulence attenuation in various pathogens. We still keep the structure information here because we think it is necessary to provide the readers this background knowledge to ease their understanding of our interpretation of the HpCtpA structure.

Minor concerns:

- 1. The sequence of a typical CTP and its description in the text is not properly mentioned, moreover, the authors should pay particular attention to the use of acronyms throughout the text. Since a lot of acronyms are used throughout the manuscript the authors should properly introduce them when they are first used.*

>> We have included the sequence range of CTPs in the introduction (see line 50-51). We have also revised the introduction as suggested so that the acronyms are explained when they are first used, see line 46, 54, 55, 56, 76, 92, 93 and 94.

2. *A schematic figure describing a typical CTP might be helpful to properly describe the enzyme, its domains and the movement during functioning of the enzyme might be helpful.*

>> The schematic diagram describing a typical CTP is added as Figure 1B.

3. *The authors can also elaborate on the importance of HpCtpA (lines 80-92) to give the reader a better understanding of the need and scope of present study.*

>> The importance of HpCtpA is elaborated as requested, see line 96-99.

Results and Discussion:

Major concerns: The authors should consider major changes in this section of the manuscript. The following are the major concerns:

Structures of H. pylori CtpA:

1. *The authors never described anything about HP1076 protein? It's cellular location? What is the molecular weight of full protein and its degraded forms? Is it an adaptor protein for HpCtpA or the substrate? Also based on the CtpA's classification HpCtp belongs to CtpA or CtpB class? The authors should clarify on this.*

>> We have revised the introduction and added background of HP1076 (See line 99-101). The cellular location of HP1076 has not be known, and here we found that HP1076 is largely present in the membrane fraction (See Supplemental Fig. S1D). Since HpCtpA is an auto-activated protein, it is unlikely for HP1076 to be an adaptor protein. Rather, results from the pull down/MS experiment indicates that HP1076 is a putative substrate of HpCtpA. Refer to Supplemental Fig. S1A, peptides covering both the N-terminus and C-terminus of HP1076 were detected. However, the C-terminus 20 residues were not detected in the degraded sample of HP1076. The classification of CtpA is clarified, see line 94.

2. *How do the authors detect and validate the presence of HpCtpA and degraded HP1076 in the pull-down assay? The methods to describe how pull-down experiments were carried out is not mentioned in M&M.*

>> The presence of HpCtpA was identified in a pull-down assay using GST-Flis- HP1076 as a bait (Supplemental Fig. S1A). As mentioned in our response to comment #1, degradation of HP1076 was examined by mass spectrometry. Further activity assay using recombinant

HpCtpA showed its proteolytic activity towards HP1076 (See Supplemental Fig. S1B). We have revised M&M and added the methodology of the pull-down assay (see line 338-344).

3. *Ultracentrifugation results also show that another population of 474 kD is observed, authors should explain this.*

>> The 474 kDa population might be protein impurity that could not be removed by the purification process or protein aggregate of CtpA during the ultracentrifugation. This 474kDa population only occupied a small proportion (6.3%), and we did not observe any other higher order oligomeric form apart from the hexamer under TEM. We have included this information in the revised manuscript (see line 125-128).

4. *The authors have mentioned that HpCtpA recombinant construct lacks first 20aa, which are probably signal peptide but didn't mention the function of signal peptide? The rationale for making CtpA Δ N46 mutant which they are using throughout this study. Why is CtpAS300A an inactive mutant? Why α - is mentioned against the names of each protein in Fig S1D? Based on WB in Fig S1D, both protein HpCtpA and HP1076 seem to be present both in membrane and cytosolic fractions, what is the explanation for these observations?*

>> In bacteria, signal peptide sequence located at the N-terminus is for protein sorting and translocation. It is a common practice to remove the signal peptide sequence in the construct for recombinant protein expression. This helps the intracellular expression of the recombinant protein with a proper folding. In our early trials, the recombinant full-length CtpA was unable to be detected in the *E. coli* cell lysate, which demonstrated the necessity to remove the signal peptide.

The reason we constructed CtpA Δ N46 was because after screening, it was the most stable truncated mutant with the longest N-terminal end retained.

According to the sequence alignment of CTPs, serine at residue 300 of HpCtpA belongs to the catalytic triad. Therefore, substitution of Ser-300 with alanine will lead to an inactive mutant. We have revised the text to clarify this (see line 118 and 119).

In Supplemental Fig. S1D, “ α –” was used to indicate antibodies against specific proteins. We have revised the labels and replaced “ α ” with “anti” to clarify this point.

We agree that expression of the two proteins, especially HP1076 were also detected in the cytoplasmic/periplasmic (C/P) fraction. In our previous study, HP1076 serves as a co-chaperone of FliS (a cytoplasmic chaperone for flagellar synthesis), the localization of HP1076 in the C/P fraction may relate to its role in cytoplasmic. For the localization of HpCtpA, it is predominately localized in the membrane fraction. The very weak band in the C/P fraction could be due to contamination carried from the membrane fraction.

5. *It seems like MST is used to derive the K_d between the inactive CtpA_{S300A} and HP1076, why is active protein not used for this experiment? Also MST experiment doesn't seem to reach the saturation limit of the substrate, moreover error bars are missing for the experiment.*

>> We did not use the wild-type CtpA in the MST assay because the active enzyme would keep binding HP1076 and releasing HP1076 digestion product, affecting the molecular movement of CtpA through the temperature gradient, and therefore fitting of the data to the K_d fit model.

In the MST experiment, the saturation point of the dose response curve is related to the concentration of CtpA added. Here, the highest CtpA_{S300A} concentration we could obtain was already applied. Further increase in CtpA_{S300A} concentration led to protein precipitation.

We have updated the figure with the error bars added.

6. *What is the rational of using the term “resting state” rather than inactive state as used in the field?*

>> CtpA is unable to process the substrate under two conditions: (1) when it is in the resting conformation, or (2) when mutations occur in residues of the catalytic triad. We use the term 'resting state' to specifically distinguish the two conditions, and indicate the conformation that renders CtpA inactive.

7. *The authors claim that “Based on the orientations of the PDZ domains and the presence of substrate peptides, the CtpA crystal structure contains three subunits in the resting state, two subunits in the active state and one protomer in an intermediate state”. To better demonstrate this observation, authors should define either a resting/active state monomer and then demonstrate a side-by-side comparison of the different states. This figure should be incorporated in the main figures of the paper. This will give a better understanding of the structural features of the protein and demonstrate the intramolecular movements in different states as authors have shown in supp. Figure 2 B-E. Also the figure legend for Supply Figure 2 is hard to understand.*

>> Supplemental figures S2B–E depict the conformations of the resting, intermediate, and active states. Since these panels largely overlap with the main figures (Figure 2BC) in content, we have retained them in the supplement. To enhance clarity, we have added additional labels to Figure S2 and revised its legend as suggested by the reviewer.

8. *The authors didn't provide the rational for designing CtpA_{S300A/K325A/Q329A}, CtpATM mutant. This mutant was supposed to be catalytically inactive, but the cryoEM map shows that the hexameric structure still contains both active and resting states, this observation should be explained.*

>> The rationale for designing CtpA_{TM} was mentioned in the manuscript (see line 141 and 142). We speculated that the difficulty in obtaining high-quality wild-type CtpA crystals was due to the structural flexibility induced by digestion of endogenous substrate. Therefore we generated an inactive variant CtpA_{TM}, in which the three residues in the catalytic triad were mutated in order to lock the protein structure in a fixed conformation with the substrate bound. However, as revealed from the cryoEM map, CtpA_{TM} subunits were still observed in either the active or resting state. This observation was in line with the HDX-MS data on CtpA_{TM} that the protein was in an equilibrium, and switching between the active and resting states is related to the PDZ domain and C-terminal motile loop instead of the catalytic triad. We have further revised the text to clarify the observation from the cryo-EM map (See line 195-198).

9. For CtpATM mutant the authors mentioned that “For the face with active subunits, a substrate peptide modeled with poly-alanine was assigned next to their shifted PDZ domain”. The authors should provide a figure for this statement, also if the active state in the hexamer is substrate induced that means the inherent tendency of HpCtpA is to remain in resting state?

>> The poly-alanine substrate model is depicted in Fig. 2C.

As shown by the HDX-MS experiment (see line 171-173), HpCtpA shifts between active and resting states independent of substrate binding. We speculate that the resting state is predominant due to its larger interface area between the PDZ domain and the main body.

10. The authors mentioned the structural regions involved in dimerization and those associated with transition from resting to active states, since similar movements were observed in BsCtpB and PaCtpA, the authors should elaborately compare present structure with the published structures and mention the similarities and differences between these structures.

>> We have included statements to compare the N- and C- dimerization in BsCtpB and tried to explain why BsCtpB doesn't have asymmetric conformational change. We also speculated that PaCtpA also has asymmetric conformational change with the presence of an adaptor protein. See line 229-236.

11. Figure citing the movement ML loop movement is missing.

>> The movement of the ML is indicated in Figure 2C when comparing the resting and active subunits.

Structural dynamics of CtpA

1. Doesn't the statement “It is plausible that CtpA subunits are in an equilibrium between the active and resting states in solution independent of substrate binding” classify HpCtpA to CtpB subfamily?

>> Although CtpA and CtpB share the characteristic of existing in equilibrium between active and resting states in solution regardless of substrate binding, this similarity is not the basis for their classification. Rather, the classification of CTPs are based on sequence homology.

Asymmetric conformational changes and the assembly of self-compartmentalization unit in CtpA

1. *The CryoEM structure was mentioned in the beginning of results but elaborately explained here. The authors should consider explaining the structural details in one heading to readers will have a holistic understanding of HpCtpA structure.*

>> At the beginning of results, we presented the presence of active and resting states in the CtpA crystal and cryo-EM structures (Section I). The equilibrium of the two states was supported by HDX-MS (Section II). The third section here aims to further investigating the structural basis of the coordination between the active and resting states by examining their differential arrangement from the cryo-EM structures of wild-type CtpA followed by in-depth structural analysis. We consider such presentation flow is necessary for the readers to follow and understand the logical order. Nevertheless, we thank the reviewer's suggestion.

2. *Based on structural data analysis, the authors provided plausible explanations of the presence of different states of protomers and functioning of different domains/residues in the functioning of this protein but the detailed figures explaining this text is missing. The authors should consider making figures explaining the text in lines 200-209, 218-224 and 227-233 to provide a better understanding of how this protein works.*

>>We have revised the labelling in Figure 4BC to better illustrate the findings from the structural analysis in lines 200-209, 218-224 and 227-233 (now lines 220-229, 246-252 and 254-261).

Minor Comments:

1. *The authors a lot of acronyms in the results and discussion sections also, such as N-dimer, subunit Sa, Sr, ML etc. kindly explain while explaining the structure probably in the beginning where structural introduction is provided for better readability throughout the manuscript.*

>> We have revised the manuscript as suggested so that the acronyms are explained when they are first used. See line 46, 54, 55, 56, 76, 92, 93, 94, 212, 240, 242, 244 and 246.

2. *Some of the figures are hard to understand, for example Supply Fig S2, it is hard to follow the figure captions.*

>> The figure captions of Fig. S2 have been revised as requested.

3. In some figures, especially structural figures, it is hard to see the text as it is embedded in the figure, kindly make necessary changes for better visualization.

>> The labels in Fig. 3 and Fig. 4 are revised as requested.

4. There is a lot of text where citations are missing, kindly include the appropriate references.

>> The citations are updated as requested.

Reviewer #1 (Remarks to the Author):

The authors have addressed all my concerns adequately except the most important: they must show that that HP1076 is cleaved. Without this simple experiment, I cannot accept the manuscript.

>> To address the reviewer's concern, we have conducted a time point CtpA activity assay in which CtpA was incubated with HP1076 for different lengths of time, and the reaction mixture was then analyzed by SDS-PAGE and further confirmed with Western blot analysis. The inactive CtpA_{S300A} was set up as a control in parallel. As shown in Response Fig. 1 below, the cleavage of HP1076 is time-dependent and only occurred in the presence of CtpA. At 0.25 h, HP1076 was partially cleaved (left panel, lane 6) and was completely cleaved by 3 h (left panel, lane 8). No cleaved band was observed for CtpA_{S300A} at either time point. The identity of the cleaved HP1076 was further confirmed using anti-HP1076 (right panel). These results showed that HP1076 is a substrate of CtpA.

The revised manuscript has been updated with the aforementioned findings (Line 122-126, Fig.S1B). In addition, we have revised the first paragraph in Results to better clarify how we identified CtpA and its proteolytic activity toward HP1076 (See Line 109-131). The label “degraded HP1076” has been replaced with “cleaved HP1076” in all figures to clarify the findings.

Response Fig. 1. Time point study of *in vitro* activity of wild-type CtpA and the CtpA_{S300A} mutant. The proteolytic activity of CtpA and the importance of Ser-300 were confirmed by a time-dependent activity assay (left). The cleavage of HP1076 was confirmed by western blot (right).

Reviewer #2 (Remarks to the Author):

I wish to thank the authors for addressing my questions/recommendations/concerns as well as those by the other reviewers.

The manuscript adds to the scientific knowledge of bacterial C-terminal proteases and should be accepted for publication as it now stands.

>> We thank the reviewer for the positive comments.

Reviewer #3 (Remarks to the Author):

The revised version provides a clearer presentation and better overall understanding of the data. However, a few minor issues remain that should be addressed:

Introduction referencing: The introduction still appears under-referenced. For example, lines 68–73 discuss the architecture of CTPs, but relevant references are missing.

>> Relevant references for the structural studies of CTPs are now cited (please see Line 70). We also revised the statement on line 69 to clarify the two subdomains of the protease core.

Revision of in-text citations are also as listed below.

Line 58, Citation (4) has been replaced with references (4, 11)

Line 73-74, Citation (9, 16, 17) are added

Line 79, Citation (18) has been replaced with references (9, 17, 18)

Line 82, Citation (11) has been replaced with references (11, 17)

Line 86, Citation (19) has been replaced with references (11, 19)

Line 88, Citation (16) has been replaced with references (12, 16)

Line 124-125 and 151, Citation (9, 11,12) have been added

Figure citation alignment:

Lines 68–70 correspond to Figure 1A but this is not cited in the text.

Lines 77–82 correspond to Figure 1B, but this figure is also not cited.

>> Figure 1A and Figure 1B are now cited (see Line 70 and 81, respectively)

Protein truncation screening: The authors should explicitly mention that they screened multiple N-terminal truncated versions of the protein and selected CtpA Δ N46 as the most stable construct retaining the longest N-terminal end.

>> We have revised the statement (see Line 119-122).

Catalytic triad referencing: Since Serine 300 belongs to the well-established catalytic triad of CTPs, appropriate references to previous studies should be included whenever these residues are discussed.

>> We have cited the relevant references for the catalytic triad (see Line 68-70, Line 124-125, 151).

HP1076 localization: I disagree with the authors' statement that the HP1076 band in the C/P fraction is "very weak." Based on the data, the authors can state that HP1076 is predominantly present in the membrane fraction. However, the data does not support the stronger conclusion that this protein is only membrane-localized.

>> We have revised the statement as suggested (see Line 128-130).

Figure 4B clarity: In the enlarged view of Figure 4B, the interactions and residues in the dynamic unit are difficult to discern. Making the ribbons transparent would help highlight these interactions more clearly.

>> We have revised the Figure 4B as suggested. The figure legend of Figure 4B has also been updated (Line 774-775).

Structural comparison (lines 229–237): As suggested by the text, a supplementary figure comparing the structures of *Bs*CtpB, *Hp*CtpA, and *Pa*CtpA would be very valuable to illustrate how the C-terminal region determines the oligomerization state of CTPs.

>> We have added a supplemental figure to illustrate how the C-terminal region of CTPs involves and determines the oligomerization state of CTPs (See Figure S9). In-text citation of the supplemental figures has also been updated.

Figure 4C inset and Supplementary Figure S9: The interactions and residues in the inset of Figure 4C are also difficult to visualize. It would be very helpful if the important residues mentioned in the text could be clearly highlighted in Supplementary Figure S9.

>> We have revised Figure S9 (now Figure S8). Residues involved in the interactions within the dynamic unit (Figure 4B inset) are marked with red triangles while residues involved in the interactions between ML1 and S_a/S_r (Figure 4C inset) are marked with blue triangles.

There are still several places in the manuscript where figures are not cited adequately. These should be carefully cross-checked and corrected.

>> We have corrected the missing citation of the figures. See Line 70, 81, 153, 230, 260, 267-268, 321, 345, 362-363.

Other corrections are listed below.

- 1) We have updated the residue labels in Figure 1A and Figure 4C inset.**
- 2) The methodologies for the CtpA activity assay and the immunoblotting analysis of HP1076 have been included (Line 379-387).**
- 3) Full name of HPLC has been added (Line 493-494).**
- 4) Use of ESPript to generate Figure S8 has been added in the Reference list.**
- 5) Figure legends of Supplemental Figures S1, S6, S8, S10, S11, S12, S13, S14.**